# Contact-electro-catalytic $CO_2$ reduction from ambient air

Nannan Wang[1,8], Wenbin Jiang [2,8], Jing Yang[3], Haisong Feng[3], Youbin Zheng[4], Sheng Wang[1], Bofan Li[1], Jerry Zhi Xiong Heng[1], Wai Chung Ong[1], Hui Ru TAN[2], Yong-Wei Zhang[3], Daoai Wang[5,6] ✉, Enyi Ye[2] ✉ & Zibiao Li[1,2,7] ✉

Traditional catalytic techniques often encounter obstacles in the search for sustainable solutions for converting $CO_2$ into value-added products because of their high energy consumption and expensive catalysts. Here, we introduce a contact-electro-catalysis approach for $CO_2$ reduction reaction, achieving a CO Faradaic efficiency of 96.24%. The contact-electro-catalysis is driven by a triboelectric nanogenerator consisting of electrospun polyvinylidene fluoride loaded with single Cu atoms-anchored polymeric carbon nitride (Cu-PCN) catalysts and quaternized cellulose nanofibers (CNF). Mechanistic investigation reveals that the single Cu atoms on Cu-PCN can effectively enrich electrons during contact electrification, facilitating electron transfer upon their contact with $CO_2$ adsorbed on quaternized CNF. Furthermore, the strong adsorption of $CO_2$ on quaternized CNF allows efficient $CO_2$ capture at low concentrations, thus enabling the $CO_2$ reduction reaction in the ambient air. Compared to the state-of-the-art air-based $CO_2$ reduction technologies, contact-electro-catalysis achieves a superior CO yield of 33 μmol g$^{-1}$ h$^{-1}$. This technique provides a solution for reducing airborne $CO_2$ emissions while advancing chemical sustainability strategy.

The development of effective and sustainable technologies to transform $CO_2$ into value-added products has received considerable attention due to the increased global warming by high $CO_2$ emissions[1-5]. Electrocatalytic $CO_2$ reduction transforms $CO_2$ into valuable products with high selectivity and energy efficiency, particularly in the generation of CO[6-11]. The broad range of applications for CO, ranging from fuel production to chemical synthesis, positions

this process a promising avenue for sustainable and clean energy technologies. However, traditional catalytic $CO_2$ reduction methods often require high energy consumption and expensive catalysts, which limit their practical applications[12-14]. Therefore, it is of great significance to develop efficient, economical and environmentally friendly $CO_2$ reduction methods. As a replaceable energy harvesting and conversion technology, triboelectric nanogenerators (TENGs)

[1]Institute of Sustainability for Chemicals, Energy and Environment (ISCE2), Agency for Science, Technology and Research (A*STAR), 1 Pesek Road, Jurong Island, Singapore 627833, Republic of Singapore. [2]Institute of Materials Research and Engineering (IMRE), Agency for Science, Technology and Research (A*STAR), 2 Fusionopolis Way, Innovis #08-03, Singapore 138634, Republic of Singapore. [3]Institute of High Performance Computing (IHPC), Agency for Science, Technology and Research (A*STAR), 1 Fusionopolis Way, #16-16 Connexis, Singapore 138632, Republic of Singapore. [4]Department of Electrical Engineering & Electronics, University of Liverpool, Brownlow Hill, Liverpool L69 7GJ, UK. [5]State Key Laboratory of Solid Lubrication, Lanzhou Institute of Chemical Physics, Chinese Academy of Sciences, Lanzhou 730000, China. [6]Shandong Laboratory of Yantai Advanced Materials and Green Manufacturing, Yantai 265503, China. [7]Department of Materials Science and Engineering, National University of Singapore, 9 Engineering Drive 1, Singapore 117576, Singapore. [8]These authors contributed equally: Nannan Wang, Wenbin Jiang. ✉e-mail: wangda@licp.cas.cn; yeey@imre.a-star.efu.sg; lizb@imre.a-star.edu.sg

converts mechanical energy into electrical energy by utilizing the charge difference generated by friction, thereby providing a replaceable way to achieve sustainable energy supply[15–19]. Mechanical energy-induced $CO_2$ reduction, on this basis, is a potential strategy for reducing greenhouse gas emissions. However, at normal temperature and pressure, mechanical energy is inefficient to reduce the chemically inert $CO_2$ molecules owing to a lack of $CO_2$ anchoring sites and a suitable mechanism in the electron transfer route. Despite the recent demonstration of proof-of-concept studies of employing contact-electro-catalysis to generate hydrogen peroxide and degrade organic dyes[20–24], we are still facing significant challenges in converting $CO_2$ into useful products utilizing mechanical energy.

The basic theory underlying contact-electro-catalysis in previous studies is that electron transfer during the actual physical contact with the interface, resulting in the generation of active free radicals that subsequently catalyze the synthesis of hydrogen peroxide or the breakdown of organic molecules. This free radical-mediated catalysis has apparent drawbacks as compared to interfacial catalysis processes in which electrons are directly engaged. In the free radical-mediated TENG catalysis, the mechanical energy is first converted into electrical energy, which further leads to reactive species to develop. This indirect conversion includes a number of steps, which could result in inefficiencies and possible energy losses. Furthermore, the TENG catalysis following this mechanism may have selectivity and controllability constraints. Since the generation of reactive radicals is a complicated and unpredictable process driven by a variety of circumstances, attaining high selectivity for certain catalytic products is difficult. Mechanisms in which electrons are directly engaged in catalytic processes, on the other hand, take a more direct and simplified approach, reducing energy waste. Furthermore, the mechanism of direct electron engagement can also be more precisely controlled to produce desired products with higher selectivity. Different from traditional electrocatalytic systems, the contact-electro-catalytic system does not require the formation of an electron flow cycle. Instead, it utilizes electrons temporarily stored on the tribolayer/catalyst surface, which are generated through the contact charging process, to directly participate in the catalytic process[25–29]. Nonetheless, contact-electro-catalytic $CO_2RR$ still faces grand challenges. First, protons required for $CO_2RR$ are produced from water molecules under high-humidity conditions, and the high humidity in turn reduces the electrical output of solid-solid TENGs and the efficiency of catalytic processes[30–33]. Moreover, the tribolayer of TENG lacks effective $CO_2$ adsorption and activation sites, which cannot sustain the steady progression of $CO_2RR$.

To address these issues, we present a method for contact-electric interfacial catalysis of $CO_2RR$ with high CO selectivity. Harnessing the prowess of quaternized cellulose nanofibers (CNFs) with the ability to immobilize water molecules, alongside electrospun PVDF loaded with Cu-PCN with electron-enrich capability, the developed TENG attains remarkable $CO_2RR$ at 99% RH, with a striking CO Faradaic efficiency of 96.24%. Further aided by the potent $CO_2$ adsorption abilities of quaternized CNF, the TENG could catalyzes $CO_2$ reduction even at an ambient milieu of low $CO_2$ concentration, such as air, resulting in a prolific CO production rate of 33 μmol $g^{-1}$ $h^{-1}$. A detailed examination of the mechanism reveals a crucial insight: The $CO_2$ adsorption capacity of quaternized CNF is much higher than that of Cu-PCN, indicating that the surface of quaternized CNF is the principal region for CO generation. Furthermore, during the contact charging phase, the Cu-PCN catalyst, loaded on PVDF fibers, accumulates an increased number of electrons. When $CO_2$ adsorbed on quaternized CNF interfaces with the Cu-PCN@PVDF, electrons transfer from Cu-PCN to $CO_2$ occurs under high electrical potential between the two triblayers, marking a critical juncture in the process. This unique technique provides a paradigm-shifting strategy that has the potential to drastically

reduce atmospheric $CO_2$ emissions, paving the way for an evolutionary path toward environmental sustainability.

## Results

### Device design for contact-electro-catalytic $CO_2RR$

The schematic structure of TENG for contact-electro-catalytic $CO_2RR$ is shown in Fig. 1a. As the electropositive triblayer of TENG, quaternized cellulose nanofiber (CNF) have a strong ability to adsorb $CO_2$. Furthermore, due to the abundance of hydroxyl groups, quaternized CNF can form hydrogen bonds with water molecules in high-humidity environments, allowing them to fix water molecules and participate in contact electrification, hence improving TENG electrical output performance in high-humidity conditions[34,35]. In a high-humidity environment, water molecules undergo oxidation reactions on the surface of quaternized CNF, leading to the generation of oxygen and protons. These protons then serve as intermediates, participating in the $CO_2RR$. The preparation process of quaternized CNF, physical photo and TEM image are shown in Supplementary Fig. 1 and Supplementary Note 1. The electronegative triblayer of TENG is electrospun PVDF loaded with single Cu atoms-anchored polymeric carbon nitride (Cu-PCN). Cu-PCN is supported by PVDF fibers, allowing the catalyst to make complete contact with the catalytic substrate. In addition, the combination of PCN with good piezoelectric properties[36–39] and PVDF will boost the electron density of the electronegative triblayer and accelerate the progress of $CO_2RR$. Density Functional Theory (DFT) was employed to illustrate the electron transfer process from PVDF to Cu-PCN, facilitating the establishment of an electric potential by Cu-PCN, as depicted in Supplementary Fig. 2 and Supplementary Note 2. The findings indicate a distinct electron transfer phenomenon from PVDF to Cu-PCN upon their contact. The schematic diagram of the contact-electro-catalytic reaction of $CO_2$ on Cu-PCN surface is shown in Supplementary Fig. 3. In a high-humidity environment, $CO_2$ molecules adsorb onto the surface of quaternized CNF electrodes, which sets the stage for a controlled electron transfer process when $CO_2$ interacts with electron-enriched Cu-PCN. Gradual reduction of $CO_2$ molecules leads to the formation of negatively charged intermediates, $CO_2^-$. Subsequent deoxygenation of $CO_2^-$, triggers the release of an oxygen atom, catalyzing the formation of CO molecules as the primary end product. Protons required for this reaction process are generated by the ionization of water on the positive triblayer surface. Figure 1b shows the correlation between electron consumption on the catalyst surface and current fluctuations in the external circuit of the TENG, see discussion in Supplementary Note 3 for details. The oxygen content in the product was analyzed using a zirconia oxygen analyzer as presented in Supplementary Fig. 4. One can see that an oxygen concentration of 0.85 ppm was detected after $CO_2RR$, confirming the taking place of the oxidation reaction of water molecules at the positive electrode.

The preparation process of Cu-PCN is shown in Supplementary Fig. 5 and Supplementary Note 4. In order to distribute Cu-PCN evenly on PVDF fibers, ultrasonically dispersed Cu-PCN was thoroughly mixed with PVDF solution before electrospinning. The optimal mass ratio of catalyst to PVDF is 1:100, which ensures that the catalyst will not agglomerate due to excessive concentration. The physical photo, SEM images and EDS of Cu-PCN@PVDF are shown in Fig. 1c, Supplementary Fig. 6a and Supplementary Fig. 7a-f, in contrast to the excess Cu-PCN agglomeration shown in Supplementary Fig. 6b. The inset of Fig. 1c shows the TEM image of Cu-PCN@PVDF, indicating that Cu-PCN is tightly attached to the PVDF fiber surface. The Cu-PCN with atomically dispersed Cu sites was chosen to ensure the selectivity of the catalyst. Figure 1d shows the spherical aberration-corrected scanning TEM (STEM) image of Cu-PCN. The abundant luminous dots evident in the image indicates the profoundly dispersed configuration of the single-atom copper catalyst[40–42]. This observation underscores the catalyst's existence as solitary copper atoms firmly situated on the catalyst

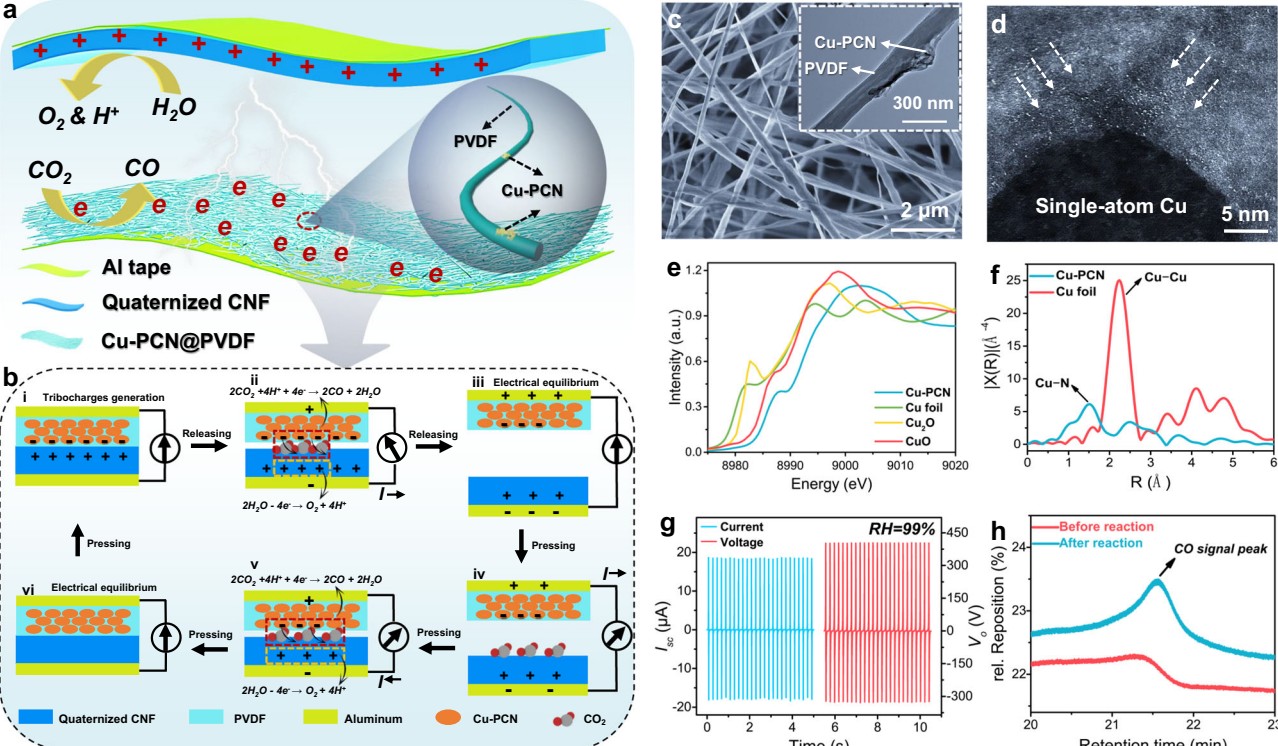

**Fig. 1 | Device design and characterizations of contact-electro-catalytic CO₂RR.**
**a** Schematic diagram showing the structure of TENG. **b** Schematic diagram of the
TENG working cycle, highlighting the integrated TENG charge generation and
catalytic charge consumption process during contact-electro-catalytic CO₂RR.
**c** SEM image of electrospun Cu-PCN@PVDF. **d** Spherical aberration-corrected
STEM image of single-atom copper in Cu-PCN. **e** Cu K-edge X-ray absorption near

edge structure (XANES) spectra of Cu-PCN and references. **f** Fourier-transformed
$k^3$-weighted Cu K-edge extended X-ray absorption fine structure (EXAFS) spectra of
Cu-PCN and the reference. **g** Output current and voltage of the TENG at 99% RH.
**h** Comparison of CO signal peaks before and after CO₂RR tested by GC. Source data
are provided as a Source Data file.

substrate. Moreover, the even distribution of these luminous points
signifies the catalyst's commendable dispersity, an attribute pivotal to
ensuring steadfast catalytic performance. To further demonstrate the
existence of single-atom copper in Cu-PCN, X-ray absorption spectra
of Cu-PCN were tested, as shown in Fig. 1e, f. The absorption edge of
Cu-PCN locates between those of CuO and Cu₂O, indicating the
coexistence of Cu⁺ (which is the predominant oxidation state) and Cu²⁺
in Cu-PCN[43]. The structural parameters were obtained by fitting the
extended X-ray absorption fine structure spectroscopy (EXAFS) spec-
tra at the Cu K-edge (Supplementary Fig. 8, Supplementary Note 5 and
Supplementary Table 1). Figure 1f shows the Fourier-transformed Cu
K-edge EXAFS spectra. The main symmetrical peak at ≈2.2 Å for Cu foil
results from the Cu-Cu bond, which cannot be observed for Cu-PCN.
Instead, a peak appeared at ≈1.6 Å indicates the formation of the Cu-N
bond in Cu-PCN.

The moisture resistance of TENG is the key to ensure the suc-
cessful progress of CO₂RR. In our previous studies, we reported the
moisture-resistant mechanism of bio-based TENG, that is, the hydroxyl
groups on the surface of the material can combine with water
molecules in the environment to participate in contact electrification
and enhance the electrical output of TENG in high-humidity
environments[44]. As a hydroxyl-rich bio-based material, quaternized
CNF also has good moisture resistance performance, as shown in
Fig. 1g and Supplementary Fig. 9. The TENG can generate a current of
18.7 μA and a voltage of 405 V at 99% RH. The electrical output of TENG
increases with humidity (Supplementary Fig. 9a and Supplementary
Note 6), and the infrared spectrum at 100 °C shows a red-shift of the
hydroxyl absorption peak of the quaternized CNF after water
absorption (Supplementary Fig. 9b), indicating the formation of
hydrogen bonds. Furthermore, the introduction of positively charged

quaternary ammonium salts[45] and PCN with piezoelectric properties
increased the electrical output performance of the quaternized CNF-
Cu-PCN@PVDF-based TENG compared with pure CNF-PVDF-based
TENG, as shown in Supplementary Fig. 10 and Supplementary Note 7.
Based on the high output under high humidity, TENG can reduce CO₂
to CO under normal conditions of pressure, temperature, and high
humidity. Comparison of CO signal peaks before and after CO₂RR
tested by Gas chromatography (GC) is shown in Fig. 1h. Before the test,
the electrical output of the TENG was accumulated for hours (4–8 h)
under pure argon atmosphere to saturate the charge (current about
18.5 μA). Then, 1 L of CO₂ gas was introduced into the airtight box. The
airtight box was filled with PTFE blocks to keep its effective volume at
5 L. A photograph of the contact-electro-catalytic CO₂RR system is
shown in Supplementary Fig. 11. Simultaneously, GC measured the
concentration of CO₂ as it was injected. At the moment when CO₂ was
just injected, the GC did not detect the signal peak of CO, indicating
that there was no CO in the airtight box. After a 5-h duration of the
reaction, GC analysis revealed a distinct peak signal of CO, suggesting
the effective reduction of CO₂ within the sealed environment, ulti-
mately yielding CO as the resulting product.

### Performance of contact-electro-catalytic CO₂RR
The Cu-PCN catalyst is essential for promoting electron transport and
accelerating the CO₂RR reaction. Various Cu amounts and catalyst
loadings were evaluated on the product yield to completely analyze
the influence of Cu-PCN on contact-electro-catalytic CO₂RR, as shown
in Fig. 2a, b. The results show that when Cu content increases, it has a
higher catalytic efficiency in CO generation. In addition, the control
TENG device (without Cu) could also generate a small quantity of CO in
contact-electro-catalytic CO₂RR. However, due to the lack of electron-

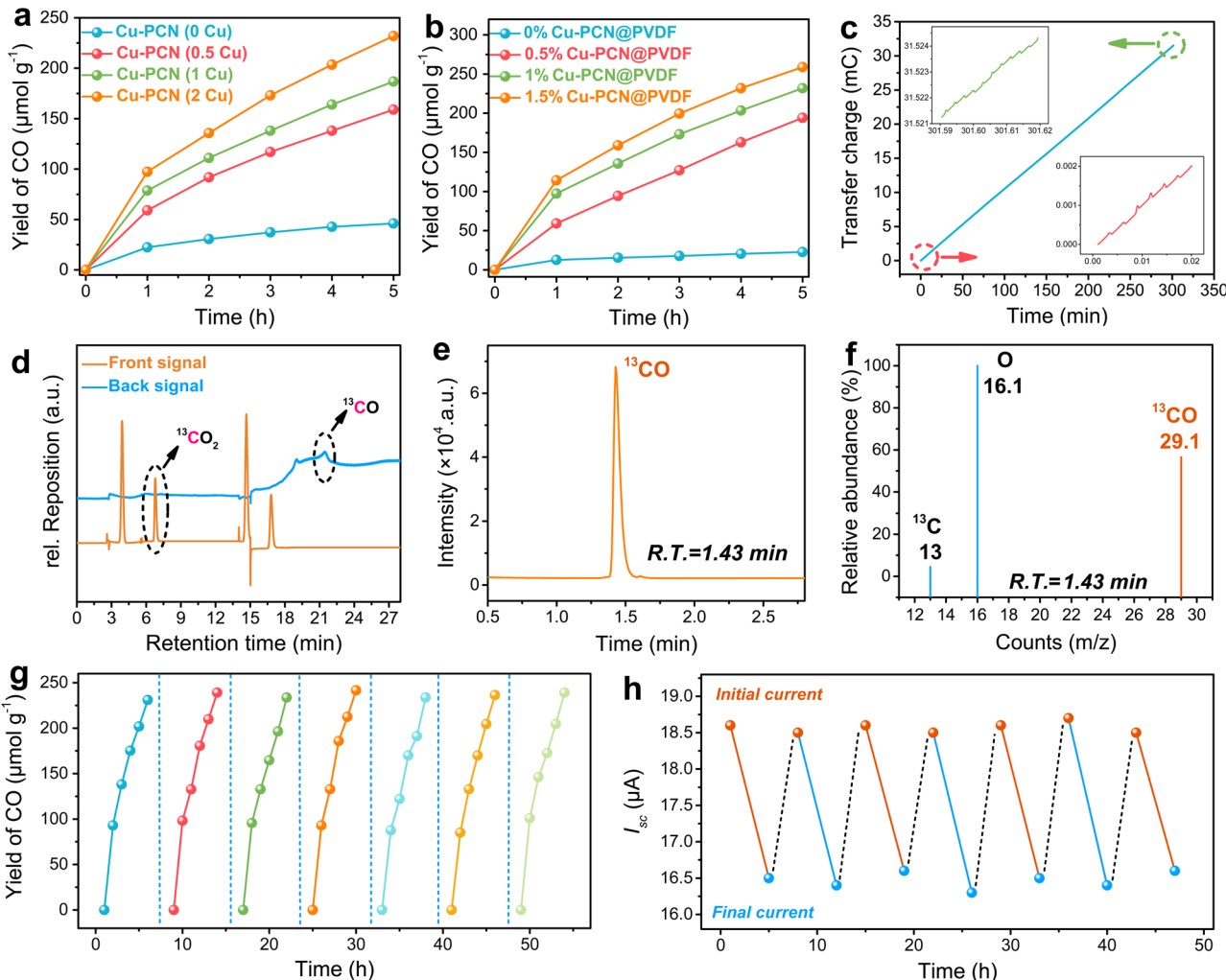

**Fig. 2 | Performance of contact-electro-catalytic CO₂RR. a** Effect of Cu content in Cu-PCN on catalytic product yield in contact-electro-catalysis within 5 h. **b** Effect of Cu-PCN content on the yield of catalytic products in contact-electro-catalysis within 5 h. **c** Calculation of transferred charges during contact-electro-catalytic CO₂RR. The inset is a magnification of the start and end of the integrated charge. **d** GC spectrum of contact-electro-catalytic CO₂RR with ¹³CO₂ as the gas source.

**e** Chromatogram of ¹³CO₂ contact-electro-catalytic reduction in selected ion monitoring (SIM) mode. **f** GC-MS of CO in the isotope experiment within ¹³CO₂ as the gas source for contact-electro-catalytic reaction. **g** Cycle runs of contact-electro-catalytic CO₂RR. **h** Changes in the output current of the TENG before and after the cyclic runs. Source data are provided as a Source Data file.

enriching effect, the electrons transferred to CO₂ will be very limited, which leads to an insignificant yield in CO production (Fig. 2a). Furthermore, the yield of CO increases with the higher loading of catalyst (Fig. 2b). It should be noted that, despite achieving the highest CO yield in the TENG device containing 1.5% of Cu-PCN loading, an excessively high catalyst content can adversely affect the formability and strength of the PVDF fibers due to the aggregation of Cu-PCN (Supplementary Fig. 6). The following studies in CO₂RR under different reaction conditions were conducted under the overall optimal performance of TENG with 1% of Cu-PCN loading in the PVDF tribolayer. For the blank experiment (0% Cu-PCN@PVDF), the CO yield was calculated using a catalyst mass of 0.68 mg (1% Cu-PCN) for a better comparison. Furthermore, a DC electric field of 410 V was established to verify the pathway of CO products, as depicted in Supplementary Fig. 12 and Supplementary Note 8. The findings revealed that there is no CO detected after 5 h test of CO₂ filled in the high voltage DC device, indicating that the CO products were generated from the contact electrocatalysis process of TENG rather than direct high-voltage dissociation. Additionally, the electric output and CO production rate of tribolayers with varying thicknesses and areas were examined, as depicted in Supplementary Fig. 13 and Supplementary

Fig. 14. One can see that the CO yield decreases with increasing thickness (Supplementary Note 9) while it is proportional to the increased area of the tribolayers (Supplementary Note 10). This is because the triboelectric charge was only generated on the surface of the TENG device, the catalyst embedded within the tribolayer will not participate the CO₂RR on the surface. However, the expanded area contributes to an increase in the TENG current, thereby positively influencing the contact-electro-catalytic CO₂RR generation (Supplementary Fig. 14). Subsequently, we calculated the number of transferred charges involved in CO₂RR, as shown in Fig. 2c. The transferred charge is obtained by real-time current integration during the CO₂RR process, and the peak shape of the transferred charge at the beginning and end of the real-time current is shown in the inset. The results show that the amount of transferred charge in the process of contact-electro-catalytic CO₂RR is 31.5 mC.

To identify the origin of CO, isotopically pure ¹³CO₂ gas was used as the source in the contact-electro-catalytic CO₂RR, in which the products were detected by GC-MS. As shown in Fig. 2d, after ¹³CO₂ was reduced for 5 h in a closed box, the product was detected by GC. The results showed that there were signal peaks of CO₂ (6.8 min retention time) and CO (21.5 min retention time) in the GC spectrum. Since the

airtight box is filled with argon gas, and the volume of injected $^{13}CO_2$ is 1 L, after the reaction, the main components of the gas in the box are Ar, $^{13}CO_2$ and $^{13}CO$. Subsequently, GC-MS was employed to further investigate the source of CO. In order to enhance the detection sensitivity of target compounds, selected ion monitoring (SIM) mode was adopted to detect the presence of $^{13}CO$ (m/z = 29) in the mixed gas. The results showed that the absorption peak of CO was detected at the retention time of 1.43 min (Fig. 2e). CO outputs three primary peaks in the mass spectrum (Fig. 2f). The strongest MS signal at m/z = 29.1 belonged to $^{13}CO$ which generates two other fragments at m/z = 13 ($^{13}C$) and m/z = 16.1 (O). These results clearly suggest that CO is derived from contact-electro-catalytic $CO_2RR$. It should be noted that O at m/z = 16.1 in the MS spectrum has a dominant signal, which may be caused by ambient oxygen carried over by the gas mixture when it was injected into the GC.

Additionally, an excellent stability in a 35-h cyclic experiment of contact-electro-catalytic $CO_2RR$ was observed in Fig. 2g. The used Cu-PCN@PVDF electrode was dried by a blower before using in each cycle of experiments. After 7 cycle runs, the yield of the reduced products remained high at about 240 μmol g$^{-1}$, suggesting a continuous production of electrons on the surface of Cu-PCN@PVDF for $CO_2RR$ under contact-separation. The output current of the TENG was recorded during each cycle run, as shown in Fig. 2h. The results show that the initial current value (saturation current) of TENG is about 18.6 μA in 7 cycles (35 h), and the current after catalytic reaction for 5 h is about 16.5 μA, which shows the stability of the electrical output during $CO_2RR$. The stability of the device was further assessed through the analysis of standard deviation in CO production and current changes during the cyclic experiments, affirming its reliability in facilitating contact electrocatalytic $CO_2RR$, as depicted in Supplementary Table 2 and Supplementary Note 11. Based on the calculation of the transferred charge for contact-electro-catalytic $CO_2RR$, we calculated the CO Faradaic efficiency ($EF_{CO}$). The specific calculation process is shown in Supplementary Table 3. The $EF_{CO}$ of contact-electro-catalytic $CO_2RR$ is calculated to be 96.24%. Compared to the $FE_{CO}$ of the state-of-the-art traditional electrocatalytic $CO_2RR$ techniques, the contact-electro-catalytic $CO_2RR$ stands out as an increased achievement among all the available references emphasizing $EF_{CO} > 90\%$ (Supplementary Fig. 15 and Table 4)[46–52].

## Mechanism of contact-electro-catalytic CO₂RR

In order to explore the mechanism of $CO_2RR$ using contact-electrocatalysis, we analyzed the adsorption performance of the materials and the catalytic process. First, the temperature programmed desorption (TPD) of $CO_2$ adsorption by quaternized CNF and Cu-PCN@PVDF was tested, as shown in Fig. 3a. The results show that Cu-PCN@PVDF only has a TCD signal peak at 78.8 °C, indicating the physical adsorption between Cu-PCN and $CO_2$ occurred. In the case of quaternized CNF, TCD signal peaks manifest at both 92.4 °C and 205 °C, highlighting robust chemical adsorption between the quaternary amino functional group and $CO_2$. This distinctly underscores the pivotal role of the quaternized CNF surface as the primary locus for $CO_2$ adsorption during the contact-electro-catalytic $CO_2RR$ process. Supplementary Fig. 16 displays the results of the thermogravimetric analysis (TGA) on both the pure and quaternized CNF membranes. According to the findings, the weight of the quaternized CNF film is reduced by 10% at a temperature of 314 °C. Therefore, the quaternized CNF will not degrade at 205 °C. Furthermore, we tested the $CO_2$ vapor adsorption of quaternized CNF and Cu-PCN@PVDF at normal temperature and pressure, as shown in Fig. 3b. We chose high concentration (20%) and low concentration (0.02%) $CO_2$ as gas sources and Ar as supplementary gas. Assessing adsorption under varying concentrations of $CO_2$ allows for a comprehensive evaluation of the $CO_2$ adsorption capabilities of both quaternary ammoniated CNF and

Cu-PCN@PVDF materials which is crucial in identifying the specific adsorption sites that play a pivotal role in the contact-electro-catalytic $CO_2RR$. The results show that when the adsorption reaches equilibrium, quaternized CNF always has a greater adsorption capacity than Cu-PCN@PVDF, regardless of $CO_2$ concentration, which is consistent with the $CO_2$-TPD test results. Furthermore, to examine the parameters affecting the $CO_2/N_2$ selectivity, Henry's law $CO_2/N_2$ selectivity was tested using TG-MS, as shown in Supplementary Fig. 17. When the gas mixture (50% $CO_2$ and 50% $N_2$) was desorbed under Ar vapor, the MS recorded the area ratio of $CO_2$ to $N_2$ to determine the $CO_2/N_2$ selectivity. The results show that the Henry's law $CO_2/N_2$ selectivity of the quaternized CNF is about 80%. Figure 3c presents a comprehensive comparison between quaternized CNF and other materials exhibiting good $CO_2$ adsorption capabilities[53–55]. These materials are ranked based on various metrics that are crucial for practical applications, encompassing adsorption capacity, adsorption rate, thermal stability, adsorption selectivity, and cycle stability. Remarkably, quaternized CNF emerges as the good performer among its counterparts. The ability to achieve strong adsorption is important to enable contact-electro-catalytic $CO_2RR$.

Subsequent experimentation delved into the impact of varying $CO_2$ concentrations on the yield of contact-electro-catalytic $CO_2RR$, as illustrated in Fig. 3d. The results indicate that, as the $CO_2$ concentration decreases from 20% to 0.02%, the yield of CO product in the contact-electro-catalytic $CO_2RR$ catalyzed by both quaternized CNF-based and pure CNF-based TENGs diminishes. The quaternized CNF-based TENG outperforms its pure CNF-based counterpart across both high and low concentration ranges. This phenomenon can be elucidated through two distinct avenues. Primarily, the TENG's current output escalates as the $CO_2$ concentration rises after reacting for 5 h, ultimately culminating in an augmented charge transfer for contact-electro-catalysis, which facilitates an upsurge in the production of CO. Secondly, in contrast to lower $CO_2$ concentrations, the surface of quaternized ammoniated CNF exhibits heightened $CO_2$ adsorption at elevated concentrations. This translates to an increased involvement of reaction substrates in the contact-electro-catalytic process, consequently leading to an augmentation in CO production. A noteworthy achievement emerges from this study-successful contact-electro-catalytic $CO_2RR$ under experimental conditions featuring even lower $CO_2$ concentrations (i.e., 0.02%) than those present in ambient air (-0.04%). This result indicates a way for further explorations into the catalytic prowess of TENG in real-world atmospheric conditions.

The output current of TENG in the contact-electro-catalytic $CO_2RR$ process will exhibit distinctive fluctuations under the influence of the two tribolayers on $CO_2$ adsorption. As shown in Fig. 3e, the output current shows an initial fall followed by a climb when there are high $CO_2$ levels present (20%). Under this condition, $CO_2$ is first adsorbed by the electropositive quaternary ammoniated CNF, leading to a decrease in current. This is owing to the electron-withdrawing effect of the carbonic acid double bond, which will withdraw electrons during the contact electrification process, resulting in a drop in the hole density of the electropositive quaternized CNF[56,57]. As the reaction proceeds, surplus $CO_2$ gradually finds residence within the interstices of PVDF electrospun fibers. This gradual accommodation imparts heightened electronegativity to the Cu-PCN@PVDF tribolayer, thereby engendering a commensurate augmentation in electrical output. However, when we eliminated the effect of the quaternary amino functional groups in the quaternized CNFs, the current showed a single changing trend, as shown in Fig. 3f. After the electropositive tribolayer quaternized CNF was replaced by pure CNF, $CO_2$ was only adsorbed on the Cu-PCN@PVDF tribolayer, which led to an increase in the electron density on the electronegative tribolayer and a consequent increase in the electrical output. Nevertheless, with a reduction in $CO_2$ concentration to 0.02%, a distinct shift in the output current pattern

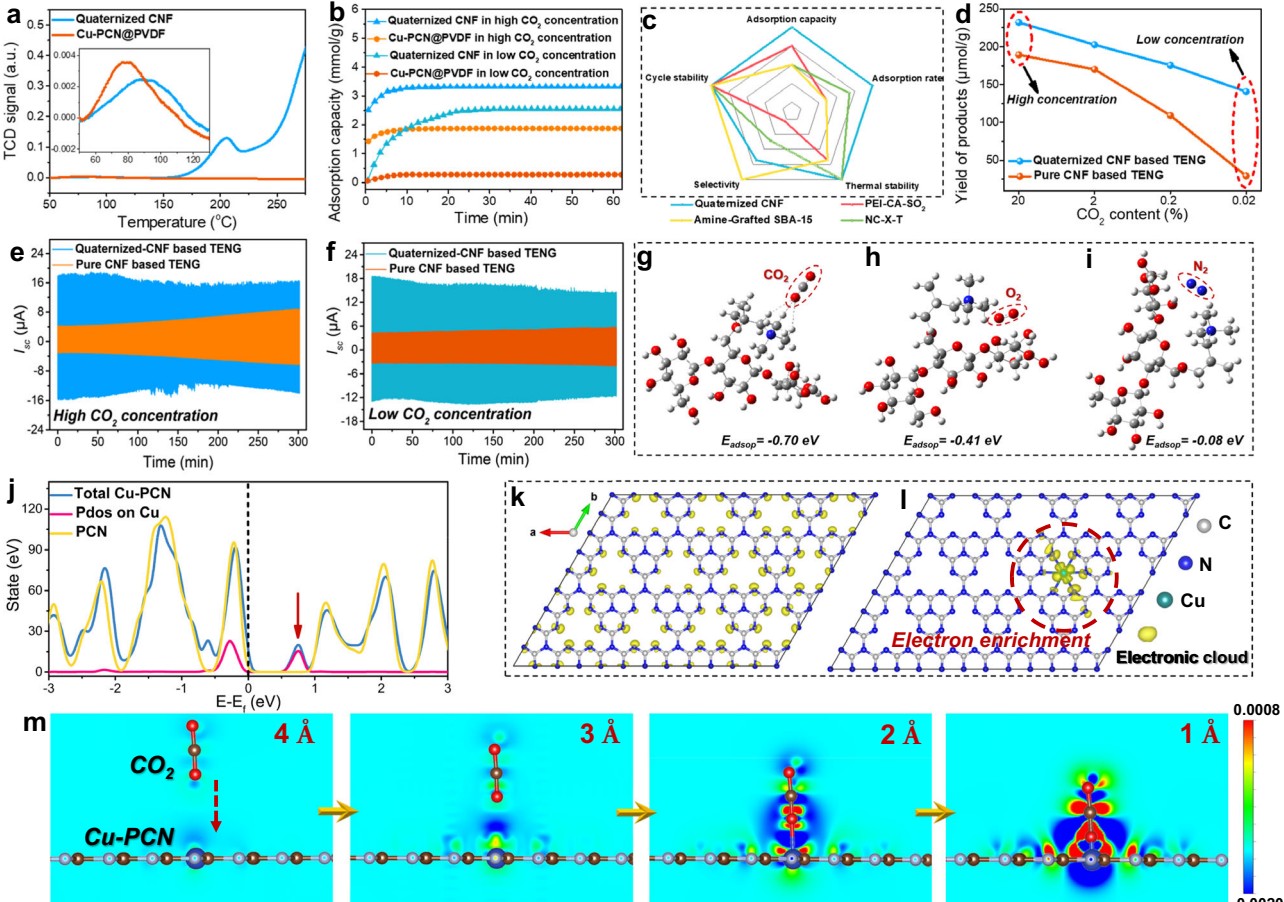

**Fig. 3 | Mechanism of contact-electro-catalytic CO₂RR. a** Comparison of CO₂-temperature programmed desorption (TPD) between quaternized CNF and Cu-PCN@PVDF. The inset is an amplification of the TCD signal in the range of 55 °C −125 °C. **b** Comparison of CO₂ vapor adsorption at room temperature and pressure of quaternized CNF and Cu-PCN@PVDF with high and low CO₂ concentrations. **c** A performance comparison of quaternized CNF with other materials that adsorb CO₂. **d** Comparison of product yields of quaternized CNF-based TENG and pure CNF-based TENG under different CO₂ concentrations for contact-electro-catalytic CO₂RR. **e, f** Comparison of the current change of contact-electro-catalytic CO₂RR of quaternary ammoniated CNF-based TENG and pure CNF-based TENG under high and low CO₂ concentrations. **g–i** Comparison of adsorption energies of quaternized CNF for CO₂, O₂ and N₂ molecules. **j** Analysis of the total density of states (TDOS) of Cu-PCN, projected density of states (PDOS) on Cu and the TDOS on PCN. **k, l** The electron distribution of the conduction band edge of PCN and Cu-PCN (iso surface = 0.004). **m** Charge distribution near Cu-PCN surface during contact. Source data are provided as a Source Data file.

emerged. Precisely, when employing the electropositive quaternized ammoniated CNF as the tribolayer in TENG, predominant adsorption takes place upon the quaternary ammoniated functional groups, leading to an ensuing continuous decrement in current. Similarly, upon substitution of quaternized CNF with pure CNF, the locus of CO₂ adsorption transitioned to Cu-PCN@PVDF, consequentially fostering an elevation in the output current. Given that the adsorbed quantity of CO₂ at diminished concentrations markedly contrasts with that at higher concentrations, a corollary emerges for pure CNF-based TENG. In this context, the surge in current at heightened concentrations surpasses the increment observed at lower concentrations.

Based on the study of CO₂ adsorption performance, we further proposed the mechanism of contact-electro-catalytic CO₂RR. During the contact electrification process of TENG, quaternized CNF effectively adsorb CO₂ molecules from the surrounding environment. Simultaneously, within the Cu-PCN@PVDF material, single-atom copper plays a crucial role in accumulating electrons and assisting electron transfer. When these two tribolayers come into contact, the electrons, accumulated by the presence of single-atom copper, which facilitate the electron transfer to the adsorbed CO₂ molecules, thereby driving and completing a catalytic reaction. To deeply study the CO₂

adsorption, DFT calculations were carried out to calculate the binding energy of CO₂ to quaternized CNF, and the results show that the adsorption energy between quaternized CNF and CO₂ molecules is 0.7 eV. Moreover, to investigate the potential competitive adsorption between CO₂, O₂, and N₂ on the quaternized CNF surface, the adsorption energies of CO₂ (-0.7 eV), O₂ (-0.41 eV) and N₂ (-0.08 eV) molecules were further calculated, as illustrated in Fig. 3g-i. The highest adsorption energy of CO₂ indicates it has the strongest adsorption on the electrode surface. Moreover, it is worth noting that the contact-electro-catalytic CO₂RR was carried out at an environmental humidity of 99% to ensure an ample supply of protons, which were essential for the reduction reaction. During the catalytic reaction, CO₂ can interact strongly with the H₂O, as suggested by the discernible acidity of the water droplet collected after the reaction (as depicted in Supplementary Fig. 18 and Supplementary Note 12). This interaction between H₂O and CO₂ has the possibility to further enhance the interaction between CO₂ and quaternized CNF. Furthermore, DFT calculations were conducted to investigate the role of electron enrichment from Cu-PCN. The analysis of the density of states (DOS) shows that pure PCN exhibits typical semiconductor characteristics, as well established by other study[58,59]. Upon the introduction of the Cu

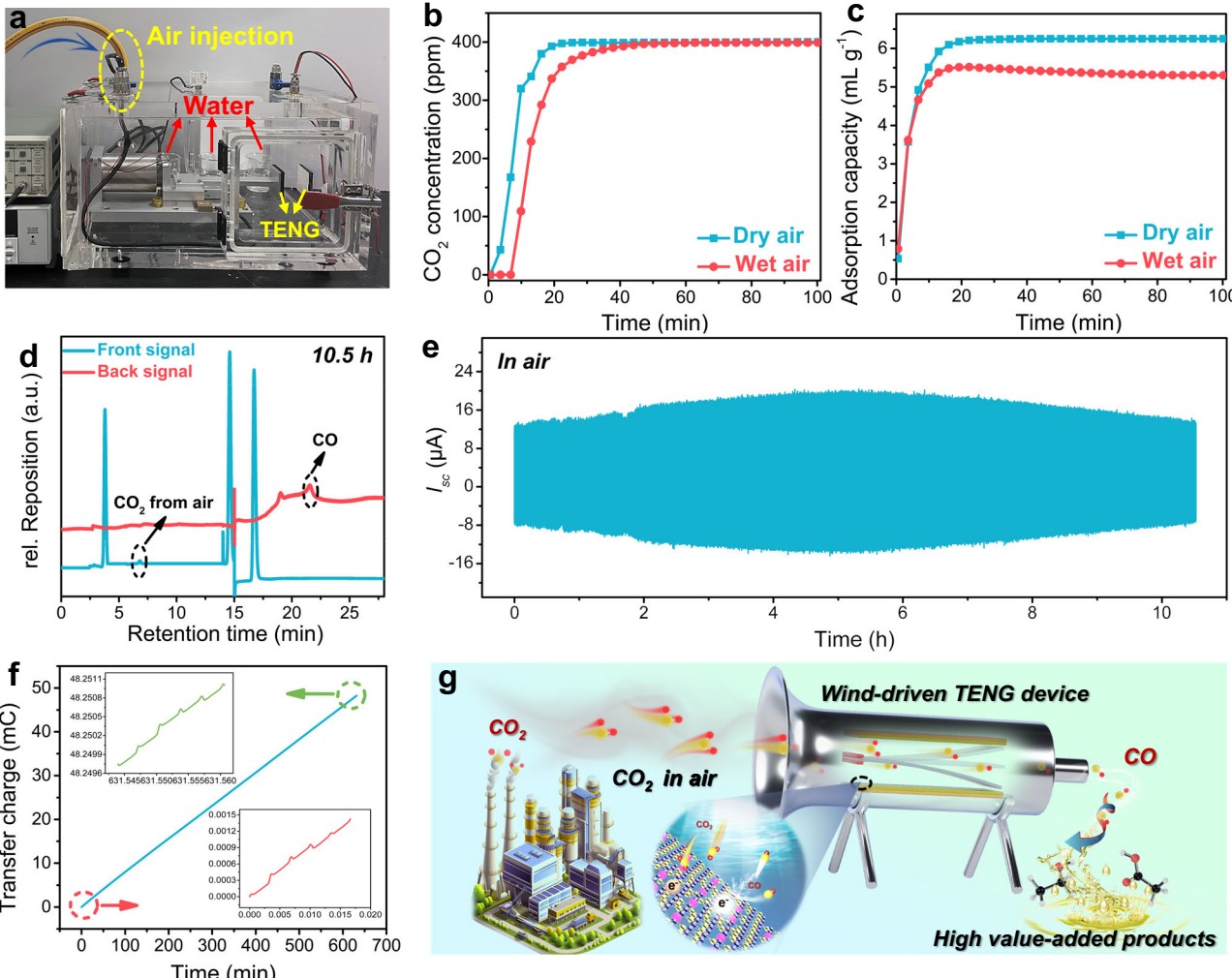

**Fig. 4 | Contact-electro-catalytic CO₂RR from ambient air. a** Photograph of the device for contact-electro-catalytic CO₂RR in a closed box at 99% RH. **b** CO₂ breakthrough curves of quaternized CNF under dry and wet (99% RH) ambient air feed at 25 °C. **c** Comparison of CO₂ adsorption capacity of quaternized CNF and Cu-PCN@PVDF in dry and wet air. **d, e** GC chromatography and real-time current change of contact-electro-catalytic CO₂RR in air. **f** Calculation of transferred charges during contact-electro-catalytic CO₂RR in air. The inset is a magnification of the start and end of the integrated charge. **g** Schematic illustration of contact-electro-catalytic CO₂ reduction in air. Source data are provided as a Source Data file.

atoms into the PCN structure, a strong hybridized peak emerges between Cu and the PCN substrate, indicating the strong interaction that stabilizes Cu on the PCN substrate. Of greater significance, the incorporation of Cu leads to the formation of a defect state near the conduction band edge (indicated by the arrow in Fig. 3j), resulting in a reduction of the band gap that will facilitate the electron injection. This defect state primarily arises from the presence of Cu, suggesting that upon injection into Cu-PCN, electrons will accumulate around the Cu atoms. Furthermore, electron distributions of PCN and Cu-PCN were computed, as depicted in Fig. 3k and l. The findings reveal that electrons in PCN are comparatively scattered, whereas electrons in Cu-PCN predominantly concentrate on copper atoms. This substantiates the electron enrichment effect attributed to the presence of single-atom copper in the Cu-PCN catalyst.

Finally, to validate the electron transfer mechanism, we conducted DFT calculations, providing insights into the charge distribution near the Cu-PCN surface during the contact process, as shown in Fig. 4m. When the separation distance between CO₂ molecules and Cu atoms exceeds 4 Å, no significant interaction is observed between the charges of CO₂ and Cu atoms on the Cu-PCN surface. However, when the distance is reduced to <4 Å, electron accumulation occurs between

the O atom of CO₂ and the Cu atom. This accumulation intensifies as the CO₂ molecule further approaches the Cu atom, leading to a noticeable build-up of electrons. Consequently, upon contact between CO₂ and Cu, electrons could swiftly transfer from the Cu atom to the CO₂ molecule. When there is no Cu atom on the PCN surface, a similar electron accumulation occurs as CO₂ molecules approach the PCN surface (Supplementary Fig. 19). However, the electron density between them is notably lower compared to the Cu-PCN interaction. As a result, when CO₂ and PCN come into contact, the efficiency of electron transfer is significantly reduced, underscoring the crucial role of single-atom copper in the electron transfer process.

## Contact-electro-catalytic CO₂RR from ambient air

As the driving force behind TENG's generation of contact-electro-catalytic charges predominantly originates from ambient mechanical energy, the contact electrically induced interfacial CO₂RR holds promise for eventual application in atmospheric settings, effecting the reduction of excess CO₂ in our environment. Moreover, the good CO₂ adsorption performance of quaternized CNF lays a strong foundation for CNF-based TENG application in catalyzing the reduction reaction of CO₂ in air. To realize contact-electro-catalytic CO₂RR in air under

experimental conditions, we used compressed air as the gas source instead of the $CO_2$/Ar mixed gas. The experiment was conducted in a confined box under normal temperature and pressure conditions with 99% humidity maintained by distilled water, as shown in Fig. 4a. First, a fixed-bed $CO_2$ breakthrough experiment was conducted on the quaternized CNF utilizing ambient air feed, which contained ~400 ppm of $CO_2$. This experiment was aimed to assess the $CO_2$ adsorption capabilities of the material within ambient air environment, as illustrated in Fig. 4b. The tests were carried out under flowing air conditions to simulate real-world conditions more realistically. The calculated fixed bed breakthrough capacities were 6.25 ml g$^{-1}$ and 5.3 ml g$^{-1}$ under dry and wet conditions, respectively (Fig. 4c). The observed breakthrough curves were steep and sharp indicating fast mass transfer in the fixed-bed. The high $CO_2$ adsorption capacities of quaternized CNF at low $CO_2$ concentrations under both dry and humid conditions indicate that quaternized CNF is a good candidate for capturing carbon dioxide from dilute gas streams (i.e., ambient air).

Based on the good $CO_2$ adsorption performance of quaternized CNF in air, quaternized CNF-Cu-PCN@PVDF-based TENG can catalyze the reduction reaction of $CO_2$ in air. After the reaction for 10.5 h, 235.65 nmol of CO is produced in the airtight box, and its GC spectrum is shown in Fig. 4d. Furthermore, real-time current changes during the reaction were recorded, as shown in Fig. 4e. The entire catalytic reaction process lasted for 10.5 h, and the current showed a trend of increasing first and then decreasing. However, when the compressed air in the system is replaced with Ar, an interesting phenomenon is observed: initially, the current tends to increase before reaching a stable level, and notably, the current does not exhibit a subsequent decrease, as shown in Supplementary Fig. 20. This observation indicates that the contact-electro-catalytic reaction takes place within the air environment, while such a reaction does not manifest when Ar is used instead. Furthermore, we calculated the transferred charges with the $CO_2$RR reaction, as shown in Fig. 4f. Similar to the method for calculating the transferred charge, the transferred charge with $CO_2$RR was obtained by current integration. The calculated transfer charge of contact-electro-catalytic $CO_2$RR in air is 48.4 mC. Subsequently, we compared the CO yield of contact-electro-catalytic $CO_2$RR in air compared with the conventional photocatalysis, as shown in Supplementary Fig. 21 and Table 5. The CO yield of contact-electro-catalytic $CO_2$RR is 33 μmol g$^{-1}$ h$^{-1}$, which is a good indicator for the yield of CO produced by catalytic $CO_2$RR in air[60–64]. Schematic illustration of contact-electro-catalytic $CO_2$RR in air is shown in Fig. 4g. In the fictional scenario, atmospheric $CO_2$ is drawn into the wind-driven TENG device. During the interaction of the TENG's quaternized CNF tribolayer with the Cu-PCN@PVDF components, mechanical energy is converted into electrical energy. This process catalyzes the reduction of atmospheric $CO_2$ to CO through the $CO_2$RR. Subsequently, the performance of the wind-driven TENG device and the impact of wind energy on CO generation during the $CO_2$RR process were comprehensively evaluated, as shown in Supplementary Fig. 22 and Supplementary Note 13. The results show that the FE$_{CO}$ of the wind-driven TENG device is ~92%, which is comparable to the FE$_{CO}$ (93.95%) obtained for the motor-driven TENG device. The slight variation in FE$_{CO}$ can be attributed to the influence of the gas flow on the $CO_2$ adsorption and product desorption process. The utilization of quaternized CNF in conjunction with the unique properties of the quaternized CNF-Cu-PCN@PVDF based TENG opens up avenues for efficient and sustainable $CO_2$ reduction within air environments, holding promise for advancing carbon capture and utilization technologies.

## Discussion

We report a stride in contact-electro-catalytic $CO_2$ reduction, offering a promising solution to the pressing challenge of sustainable carbon dioxide mitigation. Achieving a CO Faradaic efficiency exceeding 96%,

it highlights an impressive conversion of $CO_2$. Noteworthy is a good CO yield of 33 μmol g$^{-1}$ h$^{-1}$, attained even in ambient air, setting a benchmark for catalytic $CO_2$RR under low $CO_2$ conditions. The mechanism relies on the good $CO_2$ adsorption capacity of quaternized CNF, synergistically coupled with the electron-enrichment attributes of single-atom copper within Cu-PCN, which drives the contact-electro-catalytic $CO_2$RR process. This study not only advances $CO_2$ reduction technologies but also sets the path for the development of methodologies for wider contact-electro-catalytic applications.

The concept of single-atom catalysis in the $CO_2$RR field represents a cutting-edge pathway in catalysis. The seamless integration of porous coordination network technology and electrospinning enables the incorporation of different emerging single-atom catalysts into the contact-electro-catalytic system. This is poised to emerge as a universal strategy for achieving efficient catalytic platform that can power carbon dioxide reduction as well as other sustainable chemical manufacturing using industrial waste feedstock. Moreover, compared to traditional catalytic methods that typically demand substantial energy inputs, TENGs employs mechanical energy harvested from the environment to power $CO_2$RR. This not only circumvents the inherent energy consumption challenges of traditional methods but also harnesses a sustainable and environmentally friendly energy source.

Going forward, future research could focus on translating this unique contact-electro-catalytic platform into translational impact, such as incorporating it into wearables or employing it in localized carbon capture and utilization settings. These advancements could significantly contribute to mitigating $CO_2$ emissions and combating climate change. Moreover, in future investigations, through the optimization of catalyst type and the number of catalytic sites, contact-electro-catalytic $CO_2$RR holds the potential to generate products of higher value than CO, particularly in the realm of liquid products.

## Methods
### Chemical reagents
Mechanically refined cellulose nanofibril (CNF, 3.4% w/w aqueous suspension) was purchased from Tianjin Wood Elf Biotechnology Co., Ltd. Urea (CO(NH$_2$)$_2$, 99%), copper chloride (CuCl$_2$·2H$_2$O, 99.0%), polyvinylidene fluoride (PVDF, average Mw–534000), lycidyltrimethylammonium chloride (GTMAC, ≥90%), acetone (≥99.5%), N, N-Dimethylformamide (DMF, ≥99.9%), dimethyl sulfoxide (DMSO, ≥99.9%) and tetrabutylammonium hydroxide 30-hydrate (TBAH, ≥98.0%) were purchased from Sigma-Aldrich and used without further modification. DI water (18.2 MΩ) was used in all the experiments.

### Preparation of electrospun Cu-PCN@PVDF film
The preparation of Cu-PCN single-atom catalyst is shown in Supplementary Fig. 4. 0.7 g of PVDF powder was mixed with 2.9 g of acetone and stirred at room temperature. Different amounts of Cu-PCN powder (3.5 mg, 7 mg and 10.5 mg) were mixed with 1.9 g DMF and sonicated for 10 min. The Cu-PCN/DMF mixture was slowly added into PVDF/acetone, and the stirring was continued for 24 h. Furthermore, the quantity of single-atom copper in Cu-PCN can be conveniently tailored by utilizing CuCl$_2$ precursor solutions with varying concentrations. The specific preparation method is detailed in Supplementary Fig. 4. The mixture (about 4 mL) was electrospun. The parameters were: 15 KV voltage, 1 mL/h speed, 4 h,10 cm distance, and the receiving plate area is 165 cm$^2$.

### Preparation of quaternized CNF-Cu-PCN@PVDF based TENG
The preparation of quaternized CNF film is shown in Supplementary Fig. 1a. The quaternized CNF film (100 μm thickness) was cut into a size of "4 cm × 4 cm", and the aluminum electrode was pasted on the back, and copper wires were drawn out; the static-spun Cu-PCN@PVDF membrane (297 μm thickness) was cut into a size of "4 cm × 4 cm", and copper wires were drawn from the tin foil electrodes on the back. The

backs of the two electrodes were affixed with double-sided tape to form a quaternary ammoniated CNF-Cu-PCN@PVDF based TENG.

## System design of contact-electro-catalytic $CO_2RR$

Contact-electro-catalytic $CO_2RR$ was conducted within a sealed chamber fitted with a motor. Distilled water was placed inside the sealed box to maintain the ambient humidity at 99%. Prior to testing, argon gas was introduced into the sealed chamber to purge the air. Once the current from the TENG stabilizes, $CO_2$ was introduced through the air inlet, and the reaction proceeded for 5 h. At hourly intervals, a syringe was used to extract 1 mL of gas from the sealed chamber for Gas Chromatography analysis.

## Calculation of the Faradaic efficiency

The product Faradaic efficiency (FE) was calculated using the following equation:

$$FE = \frac{Z \times n \times F}{Q} \times 100\%$$

Where Z is the number of electrons transferred, n is the amount of product (mol), F is the Faraday constant (F = 96485 C/mol); Q is the transferred charge (C).

## Calculation of the CO yield

In the fabrication process of the quaternized CNF-Cu-PCN@PVDF based TENG, a precise amount of 7 mg of Cu-PCN was blended with 0.7 g of PVDF to facilitate the subsequent electrospinning procedure. The dimensions of the aluminum foil membrane, which acted as the substrate for electrospinning, were measured at "11 cm × 15 cm". Subsequently, a section of the electrospun membrane, measuring "4 cm × 4 cm", was excised to serve as the electronegative tribolayer within the TENG assembly. As a result of these preparations, the calculated weight of the Cu-PCN material amounted to 0.68 mg. As shown in Supplementary Table 5, the CO production rate in the air is 22.44 nmol h$^{-1}$, equaling a yield of 33 μmol g$^{-1}$ h$^{-1}$. Furthermore, μmol g$^{-1}$ is used to measure and compare $CO_2$ conversion capacity. The "μmol" in the unit corresponds to the CO production, whereas "g" pertains to the loading of the single-atom catalyst Cu-PCN.

## DFT calculations

All the DFT calculations were performed with the Perdew-Burke-Ernzerhof functional under the generalized gradient approximation[65,66] for the exchange-correlation interaction, as implemented in the Vienna Ab initio Simulation Package (VASP)[67,68]. A cutoff kinetic energy of 450 eV was applied to expand the electronic wave functions and the projector augmented-wave method was adopted to describe the electron-core interaction. A "3 × 3" PCN model with a Gamma-centered "3 × 3 × 1" k-mesh was used for the structural optimization. For the interaction between $CO_2$ and quaternized CNF, it was calculated with Gaussian[69]. The charge density difference of the $CO_2$/Cu-PCN system was calculated by $\Delta\rho = \rho_{total} - \rho_{CO_2} - \rho_{Cu-PCN}$, where $\rho_{total}$ is total charge density of the whole system, and $\rho_{CO_2}$ and $\rho_{Cu-PCN}$ charge densities of the $CO_2$ molecule and Cu-PCN, respectively.

## Sample characterization

The surface morphology of the samples was characterized using JEOL JSM-6710F field emission scanning electron microscope (FE-SEM). Transmission Electron Microscope (TEM) images were obtained using a TECNAI G2 TF20 high-resolution transmission electron microscope at 200 kV operating voltage (FEI, USA). At the same time, STEM-mapping images were captured with the equipped Energy Dispersive X-Ray Spectroscopy (EDX) accessory. Aberration-corrected high-angle annular dark field scanning transmission electron microscope (HAADF-STEM) measurements were conducted on a Thermo Scientific

Themis Z instrument. X-ray absorption spectroscopy (XAS) measurements were performed at the XAS beamline at the Australian Synchrotron (ANSTO) in Melbourne operated at 3.0 GeV. Liquid nitrogen cooled Si (111) double crystals were used as the double-crystal monochromator, which could cover the photon energy ranging from 5 to 19 KeV. The output voltage was measured by using NI PCIe-6259 DAQ card (National Instruments) with a load resistance of 100 MΩ, while the short-circuit current (Isc) was measured by using an SR570 low-noise current amplifier (Stanford Research System). During the electrical output test of the TENGs, an impact force of 5 N was maintained, and the frequency was 5 Hz. In order to prevent the possible influence of copper wires on $CO_2RR$, all wire clips were sealed. Fourier transform infrared spectroscopy (FTIR) spectra of CNF films were recorded on an FTIR spectrometer (Nicolet 6700, Themo Fisher Scientific Inc) from 4000 cm$^{-1}$ to 400 cm$^{-1}$ at a resolution of 4 cm$^{-1}$ with the temperature of 100 °C to ensure the free water evaporated from the surface of the quaternized CNF film. $CO_2$ adsorption/desorption measurement were performed on a TG-MS analyzer (TA Q600- HIDEN HPR 20). Before measurement, the sample was activated by heating at 200 °C for 2 h under an Ar steam. After the furnace temperature was cooled and stabilized at 35 °C, the gas was switched to the 50% $CO_2$ and 50% $N_2$ (volume percentage) mixture gas for adsorption. After 2 h, the temperature then was increased to 200 °C with a rate of 20 °C min$^{-1}$ under an Ar steam and kept for 2 h for desorption. $CO_2/N_2$ selectivity was determined according to the ratio of the integral area of $CO_2$ to that of $N_2$, recorded by MS. Thermogravimetric analyses (TGA) were carried out with a TA Q5000 (V3.15 Build 263) thermogravimetric analyzer. The test was carried out under a nitrogen atmosphere, and the temperature was raised from 40 to 600 °C at a rate of 10 °C min$^{-1}$. A zirconia oxygen analyzer (ZO-2000) with a resolution of 0.1 ppm was employed to analyze the mixed gas post-reaction. The gas from the reaction was drawn into a gas sampling bag using a micro vacuum self-priming pump, and subsequently, it was directed into the inlet of the oxygen detector. After stabilizing the flow indicator, the oxygen concentration was recorded. The products from $CO_2RR$ were analyzed by gas chromatography (GC) analysis. Gaseous products were collected from the flow gas every 1 h, and 1 mL of the collected gas was analyzed by GC (8890, Agilent) equipped with FID and TCD detectors and argon (99.999%) as the carrier gas. Isotope measurements for $CO_2RR$ were performed on an Agilent 7890 A GC-5975C MS. Selected ion monitoring (SIM) mode was used to detect the presence of $^{13}CO$ (m/z = 29) in the samples. The temperature-programmed carbon dioxide desorption ($CO_2$-TPD) was performed on a Quantachrome autosorb-iQ instrument. The sample was first pretreated in a He flow at 150 °C for 30 min, followed by cooling to 40 °C. Then, the sample adsorbed $CO_2$ for 60 min in a $CO_2$ flow. $CO_2$ desorption was conducted from 50 to 270 °C with a rate of 10 °C min$^{-1}$, and $CO_2$ was collected by a mass spectrometer. The $CO_2$ adsorption experiment at normal temperature and pressure was tested on BSD-VVS. The amount and speed of adsorption and desorption of the sample to $CO_2$ vapor with different $CO_2$ contents were determined by weighing the weight change of the sample before and after adsorption and desorption with a microbalance. The breakthrough experiments were carried out in a vertical fixed-bed reactor with ambient air feed in a down-flow manner. The concentration of the $CO_2$ in the outlet stream was monitored with a Vaisala GMP343 $CO_2$ probe (0–400 ppm of $CO_2$ is the measurement range). In wind-driven TENG-catalyzed $CO_2RR$, the flow rate of gaseous was measured using an anemometer (UT363).

## Data availability

The data supporting the findings of this study are reported in the main text or the Supplementary Information. The atomic coordinates have been deposited in the Figshare database under accession code 10.6084/m9.figshare.25920862. Source data are provided with this paper.

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

## Acknowledgements

The authors would like to thank the financial support of the Program for Taishan Scholars of Shandong Province (No. ts20190965, D.W.), the Key Research and Development Program in Shandong Province (No. SYS202203, D.W.), the Key Research Program of the Chinese Academy of Sciences (Grant No. ZDBS-ZRKJZ-TLC010, D.W.), the Western Light Project of CAS (xbzg-zdsys-202118, D.W.), Major Science and Technology Projects in Gansu Province (No. 22ZD6GA002, D.W.) and the Major Program of the Lanzhou Institute of Chemical Physics, CAS (No. ZYFZFX-5, D.W.). N.W. and Z.L. would like to thank the financial support from the Agency for Science, Technology and Research (A*STAR), Science and Engineering Research Council (SERC) Central Research Fund (Use-inspired Basic Research, Z.L.) for this work. J.Y. and Y.Z. would like to acknowledge the funding support from the Italy-Singapore Science and Technology Cooperation (Grant No. R23101R040, Y.Z.). The use of computing resources at the A*STAR Computational Centre and National Supercomputer Centre, Singapore is gratefully acknowledged.

## Author contributions

D.W, E.Y, Y.-W.Z. and Z.L. proposed the project, designed the experiments, and wrote the manuscript; N.W., W.J., H.F. and J.Y. performed the whole experiments and prepared the manuscript; S.W., Y.Z. and B.L. developed the experimental setups and assisted in analyzing the experimental data; J.H., H.T. and W.O. performed data measurements. All the authors discussed the results and commented on the manuscript.

## Competing interests

The authors declare no competing interests.
