## [Peer Review File · Nature Communications]

Contact-electro-catalytic CO₂ reduction from ambient airREVIEWER COMMENTS

Reviewer #1 (Remarks to the Author):

In this manuscript, the authors present an innovative approach to CO₂ reduction through contact-electrocatalysis using a triboelectric nanogenerator. They employ electrospun PVDF loaded with single Cu-PCN catalysts, supplemented with quaternized cellulose nanofibers (CNF) to enhance CO₂ adsorption. The authors introduce an interesting concept for CO₂ reduction under ambient conditions. However, despite commendable efforts in this study, there are several concerns that prohibit publication in Nature Communications:

1. The integration of a triboelectric nanogenerator with electrolysis for CO₂RR is indeed a noteworthy idea. However, the scenario where a TENG is attached to the sole of a shoe to convert mechanical energy into electrical energy during exercise for CO₂RR in the air presents practical limitations. The release of CO into the atmosphere without further utilization raises concerns about its impact on carbon emissions. Additionally, the device's performance appears to lag behind direct electrolysis of CO₂. Therefore, it is crucial to clarify the advantages of this device.
2. The authors claim that the triboelectric nanogenerator can produce an electric voltage of 405 V. Given the extreme conditions, it is essential to investigate whether CO₂ reduction occurs through a catalytic process or direct dissociation.
3. The authors use Cu-PCN as electrocatalysts for CO₂RR but also suggest that "the CO₂ adsorption energy of quaternized CNF is much higher than that of Cu-PCN." This raises questions about the intrinsic roles of Cu-PCN and quaternized CNF. It is important to provide experimental evidence supporting these conclusions.
4. The role of Cu-PCN as an electrocatalyst seems to be portrayed as primarily driven by the electro-enriching effect. While charge transfer is indeed important in electrocatalysis, semiconductors with limited conductivity can also act as catalysts for CO₂RR. Therefore, Cu-PCN should be discussed as a catalyst providing active binding sites for CO₂ reduction rather than solely relying on the electro-enriching effect.
5. The manuscript introduces the principle of selective CO₂ enrichment on CNF but fails to explain why O₂ and N₂ are not adsorbed on CNF. Considering the more positive redox potential of O₂/H₂O compared to CO₂/CO, it is essential to elucidate why CO₂RR in ambient conditions is favored over O₂ reduction.

Minor issues:

- In Figure 2A and B, the specific products should be clearly labeled.
- The comparison of device performance in Figure 2i lacks details; it is recommended to include at least three performance indicators: activity, selectivity, and stability.
- Evidence supporting the proposed CO₂RR mechanism is needed, particularly with regard to electron transfer from CO₂ to CO₂⁻.
- Compositions of the anodic materials in the devices (as shown in Figure S2) should be explicitly mentioned.

- Lastly, double-check the text formatting for consistency.

Reviewer #2 (Remarks to the Author):

This manuscript reports an approach of using triboelectric charges to perform CO₂ reduction. A fairly high Faradic efficiency is reported through this approach. There are a number of reports of using TENG for electrochemical applications. This appears to be the first time of using TENG for CO₂ reduction. The novelty is good. The authors conducted comprehensive experiments to confirm the mechanism and quantify the performance. The results can be impactful. However, there are still several critical issues before this manuscript can be considered for publication.

1. The mechanism is not well illustrated. Figure 1a is not illustrative and creates confusion. First, it needs to show how and where the triboelectric charges are generated and transported. There should be positive side/charge as well. It is a contact-separation process, which involves charge transfer and rebalancing. These steps should be explained clearly how the charges are transferred and balanced between electrodes, between electrode and reactants. Besides, only CO₂ is included in the schematic, what about other ions? Figure 1b shows the reaction includes H₂O, but no water molecule is given in a.
2. Electric potential on the catalyst surface is an important measure for electrochemical systems. How the electric potential is built up on the electrocatalyst surface should be illustrated and explained.
3. Only the reduction reaction of CO₂ is illustrated. This won't balance the system charge. What is the electrochemical reaction on the counter electrode? How the charge balancing is achieved in this system?
4. The PVDF and CNF fibers are not conductive. How were the charges transferred between the cathode and anode?
5. How were the transferred charges calculated in figure 2c. what were the peaks used for integration and how these peaks were measured? Again, the PVDF and CNF matrix is not conductive, it is unclear how the charge transfer can be measured externally.
6. The triboelectric effect is a surface charge effect. The mechanism seems applies to the entire volume of the electrode and catalyst. It does not make much sense that catalysts inside the fiber matrix would experience the same tribo-catalytic effect. In order to validate that, authors should show the thickness-related CO₂ conversion efficiency.
7. Micro-mol/g was used to measure and compare CO₂ conversion capacity. Please specific what weight was used and calibrated in this normalization.
8. Figure 4a is a good example. Converting CO₂ to CO into the air is not a good way to protect the environment.

Reviewer #3 (Remarks to the Author):

The manuscript, titled "Contact-electro-catalytic CO₂ reduction from ambient air," attempts to

investigate the CO₂ reduction reaction using a novel contact-electro-catalysis approach to produce carbon monoxide. While the results are interesting, they are significantly divergent from recent literature that explores the CO₂ electroreduction process in a continuous mode with innovative reactor and electrode configurations. The novelty of this approach is evident, and the manuscript shows considerable promise; however, I believe it may be better suited for publication in a journal with a more specific focus on catalysis rather than Nature Communications. In its present form, I do not recommend it for publication in Nature Communications.

To enhance the overall quality and impact of the paper, careful consideration should be given to the following comments and suggestions:

- 1) Why did the authors select carbon monoxide as the primary product of the CO₂RR process? This aspect should be highlighted in the introduction prior to outlining the manuscript's objectives.
- 2) Furthermore, it would be beneficial for the authors to provide an overview of the current state-of-the-art regarding the utilization of CNFs and PVDF loaded with Cu-PCN materials in CO₂RR applications as found in existing literature.
- 3) The methods section comprises various subsections, encompassing details about chemical reagents, the creation of electrospun Cu-PCN@PVDF film, the development of quaternized CNF-Cu-PCN@PVDF-based TENG, DFT calculations, and sample characterization. However, it lacks information regarding the experimental setup for CO₂RR applications. The authors should include a description of this setup for clarity.
- 4) What are the size and thickness of the TENG materials?
- 5) I don't fully comprehend the manuscript's structure. In the results section, it is stated that various amounts of Cu and catalyst loadings were evaluated. However, this information should be included in the methods section.
- 6) The explanation regarding the high voltage of 405 V is not sufficiently clear.
- 7) The yield of products and the current should be considered in relation to the geometric area.
- 8) I highly recommend incorporating a comprehensive explanation of the current state-of-the-art in the field of CO₂ electroreduction, with a particular focus on obtaining value-added products, especially carbon monoxide. To fortify your arguments, it is essential to cite relevant and influential references.
- 9) What is the practical implementation of these materials for the CO₂RR process, and what advantages do they offer over materials used in the literature for producing carbon monoxide?
- 10) What is the stability and durability of these materials?
- 11) Including information about the composition of the anode product stream in the literature works would be highly valuable. This data is crucial for assessing the readiness and practicality of the findings. However, the current text does not make it clear how many of the reported works in the literature actually provide this essential information.
- 12) The significance of this aspect requires more comprehensive elaboration. The author should conduct an in-depth analysis to demonstrate how their work contributes to enhancing the scalability of the CO₂ electroreduction process. Emphasizing the scalability implications will provide valuable insights into the potential practical applications and broader impact of the research. The analysis should encompass various factors, such as efficiency, cost-effectiveness, and adaptability, which are critical for evaluating the feasibility of implementing this technology on a larger scale.
- 13) An important aspect to consider in this analysis is how employing a real gaseous stream may impact the results and conclusions. The authors should address potential variations or challenges that could arise when transitioning from synthetic laboratory conditions to real-world scenarios. Discussing the

possible differences in reactant concentrations, impurities, and other environmental factors will add a layer of realism to the study and provide a more accurate assessment of the technology's viability outside controlled laboratory settings. This discussion will also contribute to a more well-rounded evaluation of the research's practical implications and potential for real-world applications, similar to the stability of the electrodes.

Comments and Authors' reply

Reviewer #1 (Remarks to the Author):

Comment: In this manuscript, the authors present an innovative approach to CO₂ reduction through contact-electrocatalysis using a triboelectric nanogenerator. They employ electrospun PVDF loaded with single Cu-PCN catalysts, supplemented with quaternized cellulose nanofibers (CNF) to enhance CO₂ adsorption. The authors introduce an interesting concept for CO₂ reduction under ambient conditions. However, despite commendable efforts in this study, there are several concerns that prohibit publication in Nature Communications:

Authors' reply: We appreciate for the reviewer's positive comments and recommendation of our manuscript for publication in Nature Communications after revision. We have revised our manuscript according to reviewer's comments point by point. Here, we list all comments and the corresponding replies as follows:

Comment 1: The integration of a triboelectric nanogenerator with electrolysis for CO₂RR is indeed a noteworthy idea. However, the scenario where a TENG is attached to the sole of a shoe to convert mechanical energy into electrical energy during exercise for CO₂RR in the air presents practical limitations. The release of CO into the atmosphere without further utilization raises concerns about its impact on carbon emissions. Additionally, the device's performance appears to lag behind direct electrolysis of CO₂. Therefore, it is crucial to clarify the advantages of this device.

Authors' reply: Thanks for the suggestion of the reviewer. We agree that attaching a triboelectric nanogenerator (TENG) to the sole of a shoe to convert mechanical energy into electrical energy during exercise for CO₂RR poses practical limitations, especially considering the uncontrolled CO emission. To mitigate the potential risk, we propose substituting human body movement (as portrayed in the original Figure 4a) with wind energy to propel the motion of the TENG's tribolayer (**Figure R1**). Following this protocol, the CO product can be systematically collected and subsequently utilized for further transformations. This adjustment not only addresses the environmental concerns

but also establishes a foundation for future improvements, such as synthesizing products with higher added value.

The CO yield in contact-electro-catalytic CO₂RR, as outlined in our manuscript, is lower than that in traditional electrocatalytic CO₂RR. This discrepancy stems from the difference in current level between the two systems, with the contact-electro-catalytic CO₂RR operating at approximately 0.02 milliamperes, while the traditional electrocatalysis typically uses a few milliamperes. With this in mind, it will be feasible to further enhance CO production in contact-electro-catalysis by improving the electricity storage performance of the friction layer or optimizing the structure of the TENG. Indeed, the distinct advantage of our device lies in the utilization of a TENG to convert mechanical energy (e.g., renewable wind energy) into sustainable electric energy, eliminating the need for external power supplies, as is the case in traditional electrocatalysis. This merit aligns well with the concept of “from nature, for nature”, positioning our device as a promising contender for sustainable and selective CO₂ conversion. Our device demonstrates high selectivity for CO₂ conversion, boasting a remarkable CO Faradaic efficiency of 96.24%. Moreover, the device can operate efficiently under ambient conditions (room temperature and air atmosphere) and has the potential for large-scale manufacturing, considering the scalable production of tribolayer materials (quaternized CNF and electrospun PVDF). This work serves as a valuable starting point for understanding the complex interaction within the contact-electro-catalysis system and provides insights that can guide future advancement in our methodology. Following the suggestion, Figure R1 has been added to the revised manuscript as Figure 4a, and the corresponding description is marked in red (Lines 5-11 of the first paragraph on page 16 of the revised manuscript).

Figure R1. Schematic representation illustrating the contact-electro-catalytic CO₂ reduction in air based on the wind-driven TENG device. Following this protocol, the CO product can be systematically collected and subsequently utilized for further transformations, such as the synthesis of products with higher added value.

Comment 2: The authors claim that the triboelectric nanogenerator can produce an electric voltage of 405 V. Given the extreme conditions, it is essential to investigate whether CO₂ reduction occurs through a catalytic process or direct dissociation.

Authors' reply: Thanks for the suggestion of the reviewer. In our manuscript, the TENG based on quaternized CNF-Cu-PCN@PVDF demonstrates the capability to generate a voltage of 405 V during operation, and notably, no electrostatic discharge was observed. To experimentally investigate the potential of the 405 V electrostatic voltage to dissociate CO₂, we arranged two fixed metal electrodes (no movement and contact during the reaction) within a sealed transparent chamber (as shown in **Figures R2a** and **R2b**). Direct current (DC) power was employed to apply a voltage of 410 V across the two iron plates, creating an electric field. As depicted in Figure R2c, no discernible peak corresponding to CO is observed in the GC spectrum after a 5-hour reaction, indicating that the CO products were generated from the contact electrocatalysis process of TENG rather than direct high-voltage dissociation. Figure R2 has been added to the revised manuscript as Figure S10, and the relevant description is marked in red (Page 8 of the revised manuscript, lines 6-11; Page 9 of the revised supporting information).

Figure R2. (a) Schematic and (b) photo of the DC electric field device of 410 V. (c) GC spectrum of the CO₂ dissociation product.

Comment 3: The authors use Cu-PCN as electrocatalysts for CO₂RR but also suggest that "the CO₂ adsorption energy of quaternized CNF is much higher than that of Cu-PCN." This raises questions about the intrinsic roles of Cu-PCN and quaternized CNF. It is important to provide experimental evidence supporting these conclusions.

Authors' reply: Thanks for the suggestion of the reviewer. In our TENG device, the synergistic interplay between quaternized CNF and the Cu-PCN catalyst collaboratively drives efficient CO₂RR. The robust CO₂ adsorption on quaternized CNF and the electron-enriching effect of Cu-PCN were demonstrated through a combination of experimental investigations and theoretical simulations. Upon contact between the two electrodes, electrons enriched on the negative electrode (Cu-PCN@PVDF) were effectively transferred to the CO₂ molecules adsorbed by quaternized CNF, facilitating the initiation of CO₂RR. To validate this inference, we conducted density functional theory (DFT) calculations on the charge distribution near the Cu-PCN surface during the catalyst-CO₂ interaction. As illustrated in **Figure R3**, there is no significant electronic interaction between CO₂ and the Cu atoms on Cu-PCN when their distance exceeds 4 angstroms. When the distance is reduced to less than 4 angstroms, electron

accumulation occurs at the region between Cu and the O atom of CO₂. This charge redistribution intensifies as the CO₂ molecule approaches closer to the Cu atom, leading to a noticeable electron build-up on CO₂. In other words, electrons can be swiftly transferred from Cu to CO₂ upon the contact of the two electrodes. Figure R3 has been added to the revised manuscript as Figure 3m, and the relevant descriptions are marked in red (Lines 1-8 of the last paragraph on page 8 of the revised manuscript).

Figure R3. Simulated charge distribution near the Cu-PCN surface during the catalyst–CO₂ interaction.

Comment 4: The role of Cu-PCN as an electrocatalyst seems to be portrayed as primarily driven by the electro-enriching effect. While charge transfer is indeed important in electrocatalysis, semiconductors with limited conductivity can also act as catalysts for CO₂RR. Therefore, Cu-PCN should be discussed as a catalyst providing active binding sites for CO₂ reduction rather than solely relying on the electro-enriching effect.

Authors' reply: Thanks for the suggestion of the reviewer. In addition to enriching electrons for promoted CO₂RR, the Cu single atoms in Cu-PCN also function as a bridge for electron transfer from Cu-PCN to CO₂. To validate the pivotal role of the Cu single atoms, we conducted a comparative analysis of the electronic catalyst–CO₂ interaction using Cu-PCN and pristine PCN as catalyst models (as illustrated in **Figures R4a** and 4b). As CO₂ molecules approach the Cu atom on Cu-PCN, the charge redistribution between Cu and the O atom of CO₂ intensifies, resulting in pronounced electron accumulation on CO₂. The efficiency of electron transfer is significantly reduced in the absence of Cu, as indicated by the reduced electron accumulation between CO₂ and pristine PCN (Figure R4b). Figure R4b has been added to the revised Supporting Information as Figure S16. All modifications are marked in red (Lines 8-15

of the last paragraph of page 8 of the revised manuscript).

Figure R4. Comparison of charge distribution near the (a) Cu-PCN surface and (b) PCN surface during the catalyst–CO₂ interaction.

Comment 5: The manuscript introduces the principle of selective CO₂ enrichment on CNF but fails to explain why O₂ and N₂ are not adsorbed on CNF. Considering the more positive redox potential of O₂/H₂O compared to CO₂/CO, it is essential to elucidate why CO₂RR in ambient conditions is favored over O₂ reduction.

Authors' reply: Thanks for the suggestion of the reviewer. In the original manuscript, our simulation results demonstrated the robust chemical adsorption of CO₂ on quaternized CNF with a notable adsorption energy of -0.70 eV. To investigate the potential competitive adsorption between CO₂, O₂, and N₂ on the quaternized CNF surface, we further calculated the adsorption energies of O₂ (-0.41 eV) and N₂ (-0.08 eV) molecules (as illustrated in **Figure R5**). The highest adsorption energy of CO₂ (-0.70 eV) indicates its strongest adsorption on the electrode surface. Moreover, it is worth noting that the contact-electro-catalytic CO₂RR was carried out at an environmental humidity of 99% to ensure an ample supply of protons essential for the reduction reaction. During the catalytic reaction, CO₂ can interact strongly with the H₂O, as suggested by the discernible acidity of the water droplet collected after the reaction (as depicted in **Figure R6**). This interaction between H₂O and CO₂ has the

possibility to further enhance the interaction between CO₂ and quaternized CNF. This disparity in adsorption abilities of the molecules (CO₂, O₂ and N₂) suggests that CO₂ can be preferentially adsorbed on the electrode surface, aligning with the observed high faradaic efficiency for CO production (FE_{CO}). However, we also observed a slightly decreased FE_{CO} from 96.24% to 93.95% when the reaction was carried out in the air atmosphere (**Figure R7**). This phenomenon can be attributed to the competitive adsorption between CO₂ and O₂ on the electrode surface. Figures R5a-R5c has been added to the revised manuscript as Figures 3g-3i. Figure R6 has been added to the revised Supporting Information as Figure S15 to demonstrate the interaction between H₂O and CO₂. All modification related descriptions are marked in red (Lines 11-21 of the second paragraph on page 13 of the revised manuscript).

Figure R5. Comparison of adsorption energies of (a) CO₂, (b) O₂ and (c) N₂ molecules on quaternized CNF.

Figure R6. (a) Photo of water droplets attached to the wall of a closed box. (b, c) Photo of the water droplets collected after the reaction. (d) pH measurement of the collected water droplets.

Figure R7. Comparison of Faradaic efficiencies for CO production during contact-electro-catalytic CO₂ reduction in CO₂ and air atmosphere.

Minor issues:

- In Figure 2A and B, the specific products should be clearly labeled.

Authors' reply: Thanks for the suggestion of the reviewer. We have annotated the presence of CO in figures depicting the product, including Figures 2a, 2b, 2g and 3f in the original manuscript. The revised figures are presented in **Figure R8**, with the corresponding figure captions highlighted in red in the revised manuscript.

Figure R8. (a) Effect of Cu content in Cu-PCN on catalytic CO yield in contact-electro-catalysis within 5 h. (b) Effect of Cu-PCN content on the yield of catalytic CO in contact-electro-catalysis within 5h. (c) Cycle runs of contact-electro-catalytic CO₂RR. (d) Comparison of CO yields of quaternized CNF based TENG and pure CNF based TENG under different CO₂ concentrations for contact-electro-catalytic CO₂RR.

- The comparison of device performance in Figure 2i lacks details; it is recommended to include at least three performance indicators: activity, selectivity, and stability.

Authors' reply: Thanks for the suggestion of the reviewer. It is noteworthy that the TENG is characterized by a substantial voltage output coupled with a modest current generation, typically in the range of tens of microamps. These characteristics are particularly advantageous in overcoming the energy barriers associated with electron transfer and molecule activation, as well as maintaining the stability of the catalyst over time. As a result, the system achieved a remarkable Faradaic efficiency of 96.24% for CO production, with negligible performance loss after at least 35 h of reaction (**Figure R9** and **Table R1**). It is also worth noting that the direct comparison of catalytic activity (CO yield) between our device and traditional electrocatalysis may not be meaningful due to the dissimilar current levels (typically several milliamperes for traditional electrocatalysis). Nonetheless, the unique advantage of our device lies in its green and sustainable energy input (mechanical energy) and potential for large-scale manufacturing. We have revisited the assessment of device performance depicted in Figure 2i in the initial manuscript. The updated analysis now encompasses additional performance metrics, specifically selectivity (Faraday efficiency) and stability. Figure R9 and Table R1 have been added to the revised manuscript as Figure 2i and Table S4, respectively, with relevant representations marked in red (Lines 10-13, 16 and 17 of the second paragraph on page 9 of the revised manuscript).

Figure R9. CO Faradaic efficiency (FE_{CO}) and catalyst duration of contact-electrocatalytic CO₂RR versus that of traditional electrocatalysis.

Table R1. FE_{CO} and FE_{CO} standard deviation of electrocatalytic and contact-electrocatalytic CO₂RR.

	N. Han et al. ¹	J. Li et al. ²	M. Liu et al. ³	X. Sun et al. ⁴	S. Zhang et al. ⁵	R. Zhao et al. ⁶	This Work
Cycle-1	82.9	90.5	94.8	94.8	94.1	99.5	96.2
Cycle-2	81.9	90	93.5	93.5	93.1	99.1	96.5
Cycle-3	83.1	90.3	95.1	93.6	92.5	99	97
Cycle-4	80.5	89	95.9	92.5	94.5	99.1	97.2
Cycle-5	83.6	90.1	95.5	92.1	94.6	98	95.1
Cycle-6	82.4	92	95.1	93.2	91.1	98.2	95.5
Cycle-7	82.5	91.7	95.3	92.8	97.2	97.5	96.2
STDEV	0.93	0.96	0.7	0.82	1.78	0.69	0.69
Average	82.4	90.5	95	93.2	93.9	98.5	96.2

• Evidence supporting the proposed CO₂RR mechanism is needed, particularly with regard to electron transfer from CO₂ to CO₂⁻.

Authors' reply: Thanks for the suggestion of the reviewer. In "Comment 3" and "Comment 4", we substantiated, through DFT calculations, that upon the contact of the electrodes, electrons can be effectively transferred from the catalyst (Cu-PCN) to CO₂ molecules adsorbed on quaternized CNF, facilitating the activation of CO₂ to CO₂⁻.

• Compositions of the anodic materials in the devices (as shown in Figure S2) should be explicitly mentioned.

Authors' reply: Thanks for the suggestion of the reviewer. We have added the components of the anode material of TENG in Figures 1a and 1b, and included the reaction process of H₂O molecules at the anode (**Figure R10a**). Furthermore, the anode composition in Figure S2 is also explicitly mentioned, as shown in Figure R10b. Figure R10b has been added to the revised Supporting Information as Figure S2, and the corresponding figure legend is marked in red (Lines 7-10 of the second paragraph on

page 4; lines 12 and 13 of the first paragraph on page 5 of the revised manuscript).

Figure R10. (a) Schematic diagram of the structure of TENG and the contact-electro-catalytic CO₂ reduction process. (b) Schematic diagram of the reaction process of CO₂ and H₂O on the electrode surfaces of TENG.

• Lastly, double-check the text formatting for consistency.

Authors' reply: Thanks for the suggestion of the reviewer. We have meticulously reviewed the text format of the manuscript and rectified any incongruities to ensure a consistent format throughout.

Reviewer #2 (Remarks to the Author):

Comment: This manuscript reports an approach of using triboelectric charges to perform CO₂ reduction. A fairly high Faradic efficiency is reported through this approach. There are a number of reports of using TENG for electrochemical applications. This appears to be the first time of using TENG for CO₂ reduction. The novelty is good. The authors conducted comprehensive experiments to confirm the mechanism and quantify the performance. The results can be impactful. However, there are still several critical issues before this manuscript can be considered for publication.

Authors' reply: We appreciate for the reviewer's positive comments and recommendation of our manuscript for publication in Nature Communications after revision. We have revised our manuscript according to reviewer's comments point by point. Here, we list all comments and the corresponding replies as follows:

Comment 1: The mechanism is not well illustrated. Figure 1a is not illustrative and creates confusion. First, it needs to show how and where the triboelectric charges are generated and transported. There should be positive side/charge as well. It is a contact-separation process, which involves charge transfer and rebalancing. These steps should be explained clearly how the charges are transferred and balanced between electrodes, between electrode and reactants. Besides, only CO₂ is included in the schematic, what about other ions? Figure 1b shows the reaction includes H₂O, but no water molecule is given in a.

Authors' reply: Thanks for the suggestion of the reviewer. We sincerely regret the inaccuracies in the depiction of the mechanism of contact-electro-catalytic CO₂RR in Figures 1a and 1b in our original manuscript. We have modified these two Figures as shown in **Figures R11a** and **R11b**. We have indicated positive and negative charges at the positive and negative electrodes of the triboelectric nanogenerator (TENG), respectively, and delineated the transfer process of electrons enriched by the catalyst during CO₂RR. Additionally, we have incorporated the oxidation process of water molecules at the positive electrode of the TENG and marked the proton transfer process. In the contact-electro-catalytic CO₂RR process, the transfer of electrons takes place

exclusively during the initial stage of contact electrification. Subsequently, electrons are transiently stored on the surface of the negative tribolayer, rather than shuttle freely between the positive and negative electrodes of the TENG^{7, 8}. These electrons can be transferred to the CO₂ adsorbed on the positive tribolayer upon contact between the two electrodes. In the meantime, positive charges are consumed by water oxidation for oxygen production, thereby maintaining a charge balance between the electrodes and reactants. A comprehensive description of this process is provided in a subsequent section. Figures R11a and R11b has been added to the revised manuscript as Figures 1a and 1b, and the relevant legends and descriptions have been marked in red (Lines 7-10 of the second paragraph on page 4 of the revised manuscript).

Figure R11. (a) Schematic diagram of the structure of TENG and the contact-electrocatalytic CO₂ reduction process. (b) Schematic diagram of the CO₂ reduction process on the Cu-PCN catalyst surface.

Comment 2: Electric potential on the catalyst surface is an important measure for electrochemical systems. How the electric potential is built up on the electrocatalyst surface should be illustrated and explained.

Authors' reply: Thanks for the suggestion of the reviewer. In the contact-electrocatalytic CO₂ reduction system, the catalyst surface potential is established through the electron transfer and electrostatic coupling of quaternized CNF and PVDF during the TENG working process. The working mechanism of TENG is shown in **Figure R12a**. When the quaternized CNF and PVDF are pressed together, the pressure bends the quaternized CNF into full contact with the PVDF, resulting in positive charges on the quaternized CNF surface and negative charges on the PVDF surface (Figure R12a (ii)). When the external pressure is reduced, the quaternized CNF partially disengages from the PVDF, and the conductive layer generates opposite charges to the tribolayer due to the electrostatic induction effect. Electrons flow from the PVDF/Al electrode to the quaternized CNF/Al electrode (Figure R12a (iii)), because the quaternized CNF/Al layer provides a negative charge and the PVDF/Al layer provides a positive charge. There is no current in the circuit until the quaternized CNF and PVDF are completely separated and the charges have achieved equilibrium (Figure R12a (iv)). Similarly, when they are pushed, a reverse current flow from the PVDF/Al electrode to the quaternized CNF/Al electrode is recorded (Figure R12a (v)). When the TENG experiences contact separation owing to the coupling of contact charging and electrostatic induction, current is produced alternately. It is noteworthy that the presence of aluminum electrodes on the back of the two tribolayers serves to collect the induced charges generated between the quaternized CNF and PVDF and the aluminum electrodes. The charges are temporarily stored on the surface of the tribolayer for catalyzing CO₂RR.⁹ In addition, the induced current is recorded and used to calculate the Faradaic efficiency of the product.

The positive and negative triboelectricity generated by the tribolayers during the contact-separation process can be explained by the electron clouds and potential well model¹⁰. As shown in Figure R12b, prior to the atomic-scale contact of the two materials, their respective electron clouds remain separated without overlap. This is the attractive force region as presented in Figure R12b. The potential well binds the electrons tightly in specific orbitals and stops them from freely escaping. When the two atoms belonging to two materials, respectively, get close to and contact with each other,

the electron clouds overlap between the two atoms to form an ionic or covalent bond. The bonding lengths are shortened even more if an external compression force is applied. In this case, the initial single potential wells become an asymmetric double-well potential, and the energy barrier between the two is lowered as a result of strong electron cloud overlap. Then electrons can then transfer from the atom of quaternized CNF to that of PVDF, resulting in contact electrification. The role played by mechanical contact of the two materials is to shorten the distance between the atoms and cause a strong overlap of their electron clouds in the repulsive region, at least in the area at which the atomic-scale contact occurs, despite the samples being larger. After contact charging occurs, electrons/holes can exist on the electronegative and electropositive tribolayers of the TENG for several hours respectively, which is the basis for establishing potential on the surface of the electrocatalyst. In order to more intuitively display the potential distribution of the PVDF tribolayer (catalyst) after contact charging, finite element simulation (FEM) is performed through COMSOL Multiphysics (Figure R12c), which is a common method to characterize the potential distribution of the tribolayer of TENG¹¹. The results show that quaternized CNF and PVDF are respectively charged with positive and negative charges with equal and opposite signs during the contact electrification process. Among them, the surface of Cu-PCN@PVDF has negative charges (electrons), which is the site of contact-electrocatalytic CO₂RR. In contrast to traditional electrocatalytic CO₂ reduction that necessitates an external power supply to consistently energize the electrodes and establish potential, contact-electro-catalysis relies on the charge stored on the surface of the tribolayer to drive the CO₂ reduction reaction. Through continuous contact-separation, charges can be continuously generated and utilized for CO₂ conversion.

Figure R12. (a) The working mechanism of TENG. (b) Electron cloud-potential well model for explaining charge transfer and release between these two materials. (c) Electric potential distribution during contact electrification of quaternized CNF and PVDF.

Comment 3: Only the reduction reaction of CO₂ is illustrated. This won't balance the system charge. What is the electrochemical reaction on the counter electrode? How the charge balancing is achieved in this system?

Authors' reply: Thanks for the suggestion of the reviewer. In our contact-electrocatalytic CO₂ reduction system, the counter electrode undergoes the electrochemical reaction of water oxidation, yielding protons and oxygen. This process is illustrated in Figure S2 of the Supporting Information in the original manuscript. To verify this process, we initially employed GC-MS to assess the oxygen production in the system, as depicted in **Figures R13a** and 13b. In comparison to the mixed gas collected before the reaction, the post-reaction gas exhibited an elevated oxygen content, suggesting the production of oxygen in the reaction system. Considering the unavoidable interference of atmospheric oxygen introduced during manual gas injection for GC-MS detection, we further carried out a more precise measurement using a zirconia oxygen analyzer (ZO-2000) with a resolution of 0.1 ppm, as illustrated in Figure R13c. For this

measurement, the gas product in the reaction system was drawn into a gas sampling bag using a micro vacuum self-priming pump, and subsequently, it was directed into the inlet of the oxygen detector. After stabilizing the flow indicator, the oxygen concentration in the mixed gas collected after the reaction was recorded and determined to be 0.85 ppm, evidencing water oxidation for oxygen production at the anode of the system. Figure R13c has been added to the modified Supporting Information as Figure S3, and the relevant description is marked in red (Lines 14-16 of the first paragraph on page 5 of the revised manuscript).

Figure R13. (a) GC and (b) MS spectra of oxygen in the mixed gas before and after the CO₂RR. (c) Detection of oxygen concentration after CO₂RR using a zirconia oxygen analyzer with a resolution of 0.1 ppm.

Comment 4: The PVDF and CNF fibers are not conductive. How were the charges transferred between the cathode and anode?

Authors' reply: Thanks for the suggestion of the reviewer. In contact-electro-catalytic CO₂RR, charge transfer exclusively takes place during the contact electrification process. This involves the transfer of electrons from PVDF to quaternized CNF, as dictated by the mechanism of contact electrification (as illustrated in Figure R12). The electrons temporarily stored on the PVDF surface can be used to propel the CO₂

reduction (**Figure R14**). In the original manuscript, the results of theoretical calculations reveal that the single-atom catalyst supported on the PVDF surface (Cu-PCN@PVDF) can effectively accumulate tribo-electrons. Therefore, the CO₂RR takes place when the CO₂ adsorbed on the surface of quaternized CNF contacts with the electrons accumulated on the Cu-PCN@PVDF. In addition, to validate this inference, we conducted DFT calculations on the charge distribution near the Cu-PCN surface during the catalyst–CO₂ interaction. As illustrated in **Figure R15**, there is no significant electronic interaction between CO₂ and the Cu atoms on Cu-PCN when their distance exceeds 4 angstroms. When the distance is reduced to less than 4 angstroms, electron accumulation occurs at the region between Cu and the O atom of CO₂. This charge redistribution intensifies as the CO₂ molecule approaches closer to the Cu atom, leading to a noticeable electron build-up on CO₂. In other words, electrons can be swiftly transferred from Cu to CO₂ upon the contact of the two electrodes. Figure R15 has been added to the revised manuscript as Figure 3m, and the relevant descriptions are marked in red (Lines 1-8 of the last paragraph on page 8 of the revised manuscript).

Figure 14. Schematic diagram of the device for contact-electro-catalytic CO₂ reduction.

Figure R15. Simulated charge distribution near the Cu-PCN surface during the catalyst–CO₂ interaction.

Comment 5: How were the transferred charges calculated in figure 2c. what were the peaks used for integration and how these peaks were measured? Again, the PVDF and CNF matrix is not conductive, it is unclear how the charge transfer can be measured externally.

Authors' reply: Thanks for the suggestion of the reviewer. The transferred charge in Figure 2c in the original manuscript was obtained by integrating the output current of the TENG. The test of the output current of the TENG is shown in **Figure R16**. Propelled by the motor, the two tribolayers of the TENG maintain a continuous cycle of contact and separation, producing charges, as illustrated in Figure R12a depicting the TENG's operational mechanism. The charges induced on the two aluminum electrodes situated at the back of the tribolayers are conveyed through copper wires to the current amplifier (SR570) and NI acquisition card. Ultimately, these charges are transformed into current signals at the computer terminal through “LabView” software. The LabView software on the computer collects the current signal, which represents the induced electricity generated by the aluminum electrode on the back of the tribolayers. The quantity of this induced charge is approximately equivalent to the number of charges generated by the tribolayers. Consequently, integrating the output current of the TENG yields a charge value that corresponds to the contact-electro-catalyzed CO₂RR.

Figure R16. Schematic diagram of the current collection process for TENGs.

Additionally, we conducted a comparison between the transferred charge measured by the Keithley 6517B electrometer during the TENG operation and the charge integrated from the output current, as shown in **Figure R17**. This verification process was undertaken to ensure the accuracy of the obtained transferred charge. This TENG is fabricated using the widely employed nylon film as the electropositive tribolayer and

the PTFE film as the electronegative tribolayer. The output current of the TENG and the transferred charge obtained by current integration are shown in Figures R17a and 17b. Within 250 s, the transferred charge of the TENG is about 2.1 μC . The transferred charge measured with the Keithley 6517B electrometer is shown in Figures R17c and 17d. The results show that the transferred charge measured by the Keithley 6517B electrometer is also about 2.1 μC , which is consistent with the charge value obtained by current integration. However, due to the limited maximum range of the Keithley 6517B electrometer, which is only 2 μC , it is unsuitable for assessing the transferred charge of a TENG operating over an extended duration (Figures R17e-R17g). This limitation arises from the continuous accumulation of transferred charge, leading to a continual increase in its value. To ensure the successful acquisition of the transferred charge during the catalysis of CO_2RR by the TENG, we utilize the integrated current for determining the transferred charge. This measurement is crucial for calculating the Faradaic efficiency of CO_2RR .

Figure R17. (a) The output current of TENG. (b) The transferred charge obtained by integrating the output current. (c) The transferred charge measured using a Keithley 6517B electrometer. (d) Real-time changes in transferred charge displayed by “LabView” software. (e-g) Measurements of the changes in transferred charge using the Keithley 6517B electrometer. (f: The maximum range of the electrometer. g: The amount of charge transferred exceeds the range.)

Comment 6: The triboelectric effect is a surface charge effect. The mechanism seems applies to the entire volume of the electrode and catalyst. It does not make much sense that catalysts inside the fiber matrix would experience the same tribo-catalytic effect. In order to validate that, authors should show the thickness-related CO₂ conversion efficiency.

Authors' reply: Thanks for the suggestion of the reviewer. To fabricate Cu-PCN@PVDF membranes with varying thicknesses, we varied the electrospinning times to 1 h, 2 h, 3 h, 4 h, and 5 h. The catalyst loading for the membranes obtained at an electrospinning time of 1 h, 2 h, 3 h, 4 h, and 5 h were determined to be 0.17 mg, 0.34 mg, 0.51 mg, 0.68 mg, and 0.85 mg, respectively. SEM results revealed a gradual increase in the thickness of the Cu-PCN@PVDF membrane with prolonged spinning time (as shown in **Figures R18a-R18f**). Then, the impact of the fiber membrane thickness on TENG output current, CO production, and yield was investigated, as shown in Figure R18g. The outcomes reveal a positive correlation between the electrical output of the TENG and the thickness of the Cu-PCN@PVDF tribolayer, escalating from 56 μm to 297 μm . However, with further thickness increments, there is a subsequent decline in the electrical output, as shown by the blue line in Figure R18g. An excessively thick tribolayer potentially impacts charge separation, consequently impeding the generation of induced charges^{12, 13}.

The electrical output of the TENG plays a pivotal role in determining the CO production during the contact-electro-catalytic CO₂RR. Consequently, as the thickness of the tribolayer increases, a trend is observed with an initial rise followed by a subsequent decline in CO production, as depicted by the green line in Figure R18g. The CO yield was also calculated by normalizing the CO production to the catalyst mass, represented by the orange line in Figure R18g. The results indicate a notable decrease in CO yield with increasing thickness of the fiber membrane. This phenomenon can be attributed to the inertness of the catalyst inside the fiber matrix, as pointed out by the reviewer. Despite the higher performance in CO yields for the thinner Cu-PCN@PVDF membrane, it is important to note that membranes that are excessively thin are

susceptible to damage upon contact and separation with quaternized CNF. Considering the long-term application perspective, the fiber membrane with an electrospinning time of 4 h was judiciously chosen as the electronegative tribolayer for TENG preparation in this study. Figure R18 has been added to the revised Supporting Information as Figure S11, and the corresponding description has been marked in red (Lines 12-16 of the first paragraph on page 8 of the revised manuscript).

Figure R18. (a) Effect of electrospinning time on the thickness of the fiber membrane. SEM cross-sectional images of the Cu-PCN@PVDF membranes obtained at various electrospinning times, including (b) 1 h, (c) 2 h, (d) 3 h, (e) 4 h, and (f) 5 h. (g) The impact of Cu-PCN@PVDF fiber membrane thickness on TENG output current (blue line), CO production (green line), and yield (orange line).

Comment 7: Micro-mol/g was used to measure and compare CO₂ conversion capacity. Please specify what weight was used and calibrated in this normalization.

Authors' reply: Thanks for the suggestion of the reviewer. The "μmol" in the unit "μmol/g" corresponds to the production of CO, whereas "g" pertains to the loading of the single-atom catalyst Cu-PCN. We have added to Table S5 in the revised Supporting Information about the weights used and calibrated for normalization in "μmol/g",

marked in red (Lines 9-12 of the description of the revised Supporting Information Table S5).

Comment 8: Figure 4a is a good example. Converting CO₂ to CO into the air is not a good way to protect the environment.

Authors' reply: Thanks for the suggestion of the reviewer. We agree that attaching a TENG to the sole of a shoe to convert mechanical energy into electrical energy for CO₂RR is not an environmentally sound practice, especially considering the uncontrolled CO emission. To address this issue, we propose substituting human body movement (as portrayed in the original Figure 4a) with wind energy to propel the motion of the TENG's tribolayer (**Figure R19**). Following this protocol, the CO product can be systematically collected and subsequently utilized for further transformations. This adjustment not only addresses the environmental concerns but also establishes a foundation for future improvements, such as synthesizing products with higher added value. This work also serves as a valuable starting point for understanding the complex interaction within the system and provides insights that can guide future advancement in our methodology. Figure R19 has been added to the revised manuscript as Figure 4a and the corresponding description is marked in red (Lines 5-11 of the first paragraph on page 16 of the revised manuscript).

Figure R19. Schematic representation illustrating the contact-electro-catalytic CO₂ reduction in air based on the wind-driven TENG device. Following this protocol, the CO product can be systematically collected and subsequently utilized for further transformations, such as the synthesis of products with higher added value.

Reviewer #3 (Remarks to the Author):

Comment: The manuscript, titled "Contact-electro-catalytic CO₂ reduction from ambient air," attempts to investigate the CO₂ reduction reaction using a novel contact-electro-catalysis approach to produce carbon monoxide. While the results are interesting, they are significantly divergent from recent literature that explores the CO₂ electroreduction process in a continuous mode with innovative reactor and electrode configurations. The novelty of this approach is evident, and the manuscript shows considerable promise; however, I believe it may be better suited for publication in a journal with a more specific focus on catalysis rather than *Nature Communications*. In its present form, I do not recommend it for publication in *Nature Communications*. To enhance the overall quality and impact of the paper, careful consideration should be given to the following comments and suggestions:

Authors' reply: Thanks for the suggestion of the reviewer. Our manuscript introduces an unprecedented approach to carbon dioxide reduction reactions that differs significantly from traditional electrocatalytic methods. This new approach utilizes triboelectric nanogenerators (TENGs) to convert mechanical energy (e.g., renewable wind energy) into sustainable electrical energy for driving CO₂RR, aligning well with the concept of "from nature, for nature". Through the incorporation of quaternized cellulose nanofibers (CNFs) and electrospun PVDF loaded with Cu-PCN, our TENG achieves remarkable CO₂ reduction even in low CO₂ concentration environments (e.g., air), with an impressive CO Faradaic efficiency exceeding 90%. The sustainable and selective CO₂ reduction achieved by contact-electro-catalysis represents a significant advancement in fundamental science. This work is believed to attract interest from wide readers and contributes significantly to the field of chemical and environmental sustainability in *Nature Communications*.

Comment 1: Why did the authors select carbon monoxide as the primary product of the CO₂RR process? This aspect should be highlighted in the introduction prior to outlining the manuscript's objectives.

Authors' reply: Thanks for the suggestion of the reviewer. Our primary motivation for

this work is to demonstrate the proof-of-concept of integrating TENG with electrolysis for CO₂ reduction. To this end, CO was chosen as the primary product because it is perhaps the most accessible product from CO₂RR and can find a wide application in industrial processes such as the Fischer–Tropsch reaction and hydroformylation¹⁴ With this consideration, our design employs Cu-PCN (highly selective for product CO) as the catalyst for CO₂RR and utilizes electronegative PVDF as the catalyst carrier for the contact-electro-catalysis of CO₂RR, aiming to produce CO. In addition, single Cu atoms-anchored PCN was chosen as the catalyst because: 1) Cu is widely used in traditional electrocatalysis¹⁵⁻¹⁹, and 2) the piezoelectric effect of PCN makes it an excellent choice for contact-electro-catalysis. While CO is the primary product in our contact-electro-catalytic CO₂RR, this does not preclude the possibility of obtaining other products with higher added value. Through the strategic modification of catalyst types and catalytic active sites, we anticipate the potential of our approach to achieve a diversified product spectrum beyond CO, such as multi-carbon species. We have emphasized this point in the introduction of the revised manuscript and cited the latest references. All changes are marked in red (Lines 3-7 of the second paragraph on page 2 of the revised manuscript).

Comment 2: Furthermore, it would be beneficial for the authors to provide an overview of the current state-of-the-art regarding the utilization of CNFs and PVDF loaded with Cu-PCN materials in CO₂RR applications as found in existing literature.

Authors' reply: Thanks for the suggestion of the reviewer. In our manuscript, we introduced the concept of integrating TENG with electrolysis (contact-electro-catalysis) for CO₂ reduction for the first time. In this novel approach, the reaction is driven by mechanical energy through a TENG, eliminating the need for external power supplies, as is the case in traditional electrocatalysis⁹. Notably, the electrode materials (PVDF and quaternized CNF) used for TENG are polymeric (insulating) and not typically suited for traditional electrocatalytic CO₂RR. Nonetheless, these materials are commonly used as the tribolayer for TENG, featured with high universality^{20, 21}. Furthermore, the combination of Cu-PCN with PVDF is because: 1) Cu is widely used

in traditional electrocatalysis, and 2) the piezoelectric effect of PCN makes it an excellent choice for contact-electro-catalysis. The synergy of these materials not only underscores the practicality of our approach but also presents a new avenue for CO₂ reduction.

Comment 3: The methods section comprises various subsections, encompassing details about chemical reagents, the creation of electrospun Cu-PCN@PVDF film, the development of quaternized CNF-Cu-PCN@PVDF-based TENG, DFT calculations, and sample characterization. However, it lacks information regarding the experimental setup for CO₂RR applications. The authors should include a description of this setup for clarity.

Authors' reply: We thank the reviewer for the valuable comments. As suggested, we have added new information to describe the CO₂RR application experimental setup in the "Methods" section in the revised manuscript and all modifications have been marked in red (the first paragraph on page 20 of the revised manuscript). The setup for CO₂RR applications has also been demonstrated by an actual photo as presented in Figure 4b.

Comment 4: What are the size and thickness of the TENG materials?

Authors' reply: Thanks for the suggestion of the reviewer. The dimensions of the two tribolayers constituting the TENG are both "4 cm×4 cm." Additionally, the thickness of the electropositive tribolayer, quaternized CNF film, is about 100 μm, while the Cu-PCN@PVDF film has a thickness of 297 μm. The cross-sectional SEM image of the Cu-PCN@PVDF film and a physical photograph of the Cu-PCN@PVDF electrode are shown in **Figures R20a** and R20b. We have added the thickness of quaternized CNF and Cu-PCN@PVDF to the "Methods" section in the revised manuscript, and the corresponding descriptions are marked in red (Lines 2 and 5 of the fourth paragraph on page 19 of the revised manuscript).

Figure R20. (a) Cross-sectional SEM image of the Cu-PCN@PVDF film. (b) Physical photograph of the Cu-PCN@PVDF electrode.

Comment 5: I don't fully comprehend the manuscript's structure. In the results section, it is stated that various amounts of Cu and catalyst loadings were evaluated. However, this information should be included in the methods section.

Authors' reply: Thanks for the suggestion of the reviewer. We apologize for the lack of information on the amount of single-atom copper and the content of the catalyst in the “Methods” section. We have added the amount of Cu in Cu-PCN and the loading of Cu-PCN to the “Methods” section, all modifications have been marked in red (Lines 3 and 4 of the third paragraph on page 19 of the revised manuscript).

Comment 6: The explanation regarding the high voltage of 405 V is not sufficiently clear.

Authors' reply: Thanks for the suggestion of the reviewer. We apologize for any confusion resulting from our insufficient clarification regarding the 405 V voltage in the context of contact-electro-catalytic CO₂RR. In contact-electro-catalytic CO₂RR, the term '405 V' denotes the voltage generated by the TENG, corresponding to the potential difference between the two charged tribolayers of the TENG²². The catalyst surface potential is established through the electron transfer and electrostatic coupling of quaternized CNF and PVDF. The working mechanism of TENG is shown in **Figure R21a**. When the quaternized CNF and PVDF are pressed together, the pressure bends the quaternized CNF into full contact with the PVDF, resulting in positive charges on the quaternized CNF surface and negative charges on the PVDF surface (Figure R21a

(ii). When the external pressure is reduced, the quaternized CNF partially disengages from the PVDF, and the conductive layer generates opposite charges to the tribolayer due to the electrostatic induction effect. Electrons flow from the PVDF/Al electrode to the quaternized CNF/Al electrode (Figure R21a (iii)), because the quaternized CNF/Al layer provides a negative charge and the PVDF/Al layer provides a positive charge. There is no current in the circuit until the quaternized CNF and PVDF are completely separated and the charges have achieved equilibrium (Figure R21a (iv)). Similarly, when they are pushed, a reverse current flow from the PVDF/Al electrode to the quaternized CNF/Al electrode is recorded (Figure R21a (v)). When the TENG experiences contact separation owing to the coupling of contact charging and electrostatic induction, current is produced alternately. It is noteworthy that the presence of aluminum electrodes on the back of the two tribolayers serves to collect the induced charges generated between the quaternized CNF and PVDF and the aluminum electrodes. The tribo-charges are temporarily stored on the surface of the tribolayer for catalyzing CO₂RR. In addition, the induced current is recorded and used to calculate the Faradaic efficiency of the product. In order to more intuitively display the potential distribution of the PVDF tribolayer (catalyst) after contact charging, finite element simulation (FEM) is performed through COMSOL Multiphysics (Figure R21b), which is a common method to characterize the potential distribution of the tribolayer of TENG. The results show that quaternized CNF and PVDF are respectively charged with positive and negative charges with equal and opposite signs during the contact electrification process. Among them, the surface of Cu-PCN@PVDF has negative charges (electrons), which is the site of contact-electro catalytic CO₂RR.

Figure R21. (a) The working mechanism of TENG. (b) Electric potential distribution during contact electrification of quaternized CNF and PVDF.

Comment 7: The yield of products and the current should be considered in relation to the geometric area.

Authors' reply: Thanks for the suggestion of the reviewer. We conducted a comparative analysis of the output current and CO production in TENG with electrode geometric areas of "2 cm×2 cm," "3 cm×3 cm," "4 cm×4 cm," and "5 cm×5 cm," as illustrated in **Figures R22a** and R22b. It turns out that both the current output and the CO production correlate with the geometric area of the TENG. Larger geometric areas lead to enhanced generation of frictional charges, thus facilitating higher electrical output and CO production^{23, 24}. In prior studies demonstrating the electrical output of TENGs²⁵⁻²⁷, researchers commonly maintained the electrode size at "4 cm × 4 cm." In our manuscript, we have opted to fabricate TENGs with an electrode area of "4 cm × 4 cm" to provide a more direct and illustrative representation of the TENG's electrical output performance. Figure R22 has been added in the revised Supporting Information as Figure S12 with the corresponding description marked in red (Lines 16-18 of the first paragraph on page 8 of the revised manuscript).

Figure 22. (a) Comparison of the current output and CO yield of TENGs with different geometric areas. (b) Entity photographs of Cu-PCN@PVDF electrodes with different geometric areas.

Comment 8: I highly recommend incorporating a comprehensive explanation of the current state-of-the-art in the field of CO₂ electroreduction, with a particular focus on obtaining value-added products, especially carbon monoxide. To fortify your arguments, it is essential to cite relevant and influential references.

Authors' reply: Thank you for your valuable feedback and insightful suggestions. We appreciate your recommendation to provide a comprehensive explanation of the current

state-of-the-art in the field of CO₂ electroreduction, particularly emphasizing the production of value-added products such as CO. Indeed, conventional electrocatalytic CO₂ reduction has demonstrated significant prowess in generating a variety of valuable products, with CO being a particularly versatile and economically relevant compound. CO finds utility in various applications²⁸⁻³³, including as a chemical feedstock for industrial synthesis (such as Fischer–Tropsch synthesis and methanol production)^{14, 34, 35} and as a key intermediate for the production of valuable chemicals³⁶⁻³⁸. However, the efficiency of traditional electrocatalysis is hindered by high energy consumption and the costliness of catalysts, prompting the emergence of alternative approaches like contact-electro-catalysis for CO₂ reduction. In contrast to traditional methods, contact-electro-catalysis offers a promising avenue by harnessing mechanical energy through TENGs. This strategy not only addresses the energy consumption challenges but also introduces an innovative pathway for CO₂ reduction. The unique method we propose, utilizing quaternized CNF and electrospun PVDF loaded with Cu-PCN, achieves exceptional CO₂ reduction efficiency, even in low CO₂ concentration environments. The combination of these materials not only demonstrates the feasibility of our approach but also showcases the potential for highly selective CO₂ conversion. By highlighting the limitations of traditional electrocatalysis and underscoring the advantages of contact-electro-catalysis in terms of energy efficiency and cost-effectiveness, our manuscript aims to contribute to the ongoing discourse on sustainable CO₂ reduction. We will certainly fortify our arguments by incorporating relevant and influential references to provide a more thorough overview of the current state-of-the-art in the field. In the revised manuscript, we have enhanced the 'Introduction' section by incorporating a comprehensive overview of the current state-of-the-art technology, highlighting the multifaceted applications of the CO product. Citations from current influential literature have been included, and all modifications are marked in red (Lines 3-7 of the second paragraph on page 2 of the revised manuscript).

Comment 9: What is the practical implementation of these materials for the CO₂RR process, and what advantages do they offer over materials used in the literature for

producing carbon monoxide?

Authors' reply: Thanks for the suggestion of the reviewer. In the manuscript, we introduced the concept of integrating TENGs with electrolysis (contact-electrocatalysis) for CO₂ reduction for the first time. Unlike traditional electrocatalysis, contact-electro-catalysis does not necessitate an external power supply to provide electrons for CO₂ reduction. Both the tribolayer materials used to assemble the TENG device (quaternized CNF and PVDF) are insulating, rendering them unsuitable for direct use in electrocatalytic CO₂RR. Nonetheless, these materials are commonly used as the tribolayer for TENG, featured with high universality. In detail, quaternized CNF featuring a hydroxyl-rich nature is a commonly used material in the fabrication of moisture-resistant TENG^{25,39}, ensuring the supplement of water molecules and protons for CO₂RR. The robust adsorption of CO₂ on quaternized CNF was also confirmed by our experimental investigations and theoretical simulations. PVDF stands out as one of the most commonly used negatively charged tribolayer materials for TENG preparation owing to its excellent piezoelectric performance and hydrophobicity²¹. Furthermore, its exceptional electrospinning capability⁴⁰ makes PVDF an optimal support for catalyst loading (Cu-PCN). The piezoelectric effect of Cu-PCN further enhances the electrical output performance of TENG, proving beneficial for contact-electro-catalytic CO₂RR. The Cu-PCN with Cu single atoms as active sites for CO₂ reduction holds immense potential in achieving high catalytic activity, product selectivity, and catalyst stability⁴¹⁻⁴⁵. The effective combination of quaternized CNF, PVDF, and Cu-PCN, along with the rational design of the TENG device, ultimately contributes to the success of our contact-electro-catalytic CO₂RR, presenting a new avenue for CO₂ conversion.

Comment 10: What is the stability and durability of these materials?

Authors' reply: Thanks for the suggestion of the reviewer. In the original manuscript, we conducted a cyclic experimental evaluation of CO₂ reduction to examine the material and device durability, with a focus on the associated current variations before and after the test, as illustrated in **Figure R23**. The findings indicated that throughout the 7-cycle experiment, the CO production remained consistently around 240 μmol g⁻¹,

and there were negligible fluctuations in the current before and after the reaction. After nearly 35 h of testing, no apparent damage to the tribolayers (quaternized CNF and PVDF) of the TENG and detachment of the catalyst (Cu-PCN) were observed. These observations underscore the high stability and durability of the materials and device. To further investigate the stability of materials exposed to electrocatalytic CO₂RR, we analyzed the standard deviation (STDEV) of CO production and current changes during the cycling experiment, as shown in **Table R2**. The summary of the Faradaic efficiency for CO production (FE_{CO}) and its standard deviation in electrocatalytic and contact-electro-catalytic CO₂RR is presented in **Table R3**. The outcomes reveal a standard deviation of 3.28 μmol g⁻¹ for CO production, signifying a minimal degree of dispersion. In addition, the standard deviations of the TENG current are about 0.07 μA and 0.1 μA before and after the reaction, respectively, indicating low current fluctuations. These results underscore the high stability of the materials employed for contact-electrocatalytic CO₂RR. Tables R2 and R3 have been added to the modified supporting information as Tables S2 and S4, and descriptions related to the stability and durability of the material are marked in red (Lines 10-13 of the second paragraph on page 9 of the revised manuscript).

Figure R23. (a) Cycle runs of contact-electro-catalytic CO₂RR. (b) Changes in the output current of the TENG before and after each reaction during the cyclic runs.

Table R2. Analyzing standard deviation in CO yield and TENG current across 7 experimental cycles.

Cycles	1	2	3	4	5	6	7	Standard deviation
Yield	239	231	239	234	236	241	234	3.28

(μmolg^{-1})								
I_{sc1} (μA)	18.6	18.5	18.6	18.5	18.6	18.7	18.5	0.07
I_{sc2} (μA)	16.5	16.4	16.6	16.3	16.5	16.4	16.6	0.10

Table R3. FE_{CO} and FE_{CO} standard deviation of electrocatalytic and contact-electrocatalytic CO₂RR.

	N. Han et al. ¹	J. Li et al. ²	M. Liu et al. ³	X. Sun et al. ⁴	S. Zhang et al. ⁵	R. Zhao et al. ⁶	This Work
Cycle-1	82.9	90.5	94.8	94.8	94.1	99.5	96.2
Cycle-2	81.9	90	93.5	93.5	93.1	99.1	96.5
Cycle-3	83.1	90.3	95.1	93.6	92.5	99	97
Cycle-4	80.5	89	95.9	92.5	94.5	99.1	97.2
Cycle-5	83.6	90.1	95.5	92.1	94.6	98	95.1
Cycle-6	82.4	92	95.1	93.2	91.1	98.2	95.5
Cycle-7	82.5	91.7	95.3	92.8	97.2	97.5	96.2
STDEV	0.93	0.96	0.7	0.82	1.78	0.69	0.69
Average	82.4	90.5	95	93.2	93.9	98.5	96.2

Comment 11: Including information about the composition of the anode product stream in the literature works would be highly valuable. This data is crucial for assessing the readiness and practicality of the findings. However, the current text does not make it clear how many of the reported works in the literature actually provide this essential information.

Authors' reply: Authors' Response: Thank you for the reviewer's insightful suggestions. In our contact-electro-catalytic CO₂ reduction system, the counter electrode undergoes the electrochemical reaction of water oxidation, yielding protons and oxygen. This process is illustrated in Figure S2 of the Supporting Information in the original manuscript. To verify this process, we initially employed GC-MS to assess the oxygen production in the system, as depicted in **Figures R24a** and **R24b**. In

comparison to the mixed gas collected before the reaction, the post-reaction gas exhibited an elevated oxygen content, suggesting the production of oxygen in the reaction system. Considering the unavoidable interference of atmospheric oxygen introduced during manual gas injection for GC-MS detection, we further carried out a more precise measurement using a zirconia oxygen analyzer (ZO-2000) with a resolution of 0.1 ppm, as illustrated in Figure R24c. For this measurement, the gas product in the reaction system was drawn into a gas sampling bag using a micro vacuum self-priming pump, and subsequently, it was directed into the inlet of the oxygen detector. After stabilizing the flow indicator, the oxygen concentration in the mixed gas collected after the reaction was recorded and determined to be 0.85 ppm, evidencing water oxidation for oxygen production at the anode of the system. Figure R24c has been integrated into the modified Supporting Information as Figure S3, with the relevant description marked in red (Lines 14-16 of the first paragraph on page 5 of the revised manuscript).

Figure R24. (a) GC and (b) MS spectra of oxygen in the mixed gas before and after the CO₂RR. (c) Detection of oxygen concentration after CO₂RR using a zirconia oxygen analyzer with a resolution of 0.1 ppm.

Comment 12: The significance of this aspect requires more comprehensive elaboration. The author should conduct an in-depth analysis to demonstrate how their work contributes to enhancing the scalability of the CO₂ electroreduction process. Emphasizing the scalability implications will provide valuable insights into the potential practical applications and broader impact of the research. The analysis should encompass various factors, such as efficiency, cost-effectiveness, and adaptability, which are critical for evaluating the feasibility of implementing this technology on a larger scale.

Authors' reply: Thank you for your insightful feedback and suggestion to provide a more comprehensive elaboration on the significance of our work, particularly in terms of enhancing the scalability of the CO₂ electroreduction process. We have merged the reviewer's comment with the subsequent one to offer a more comprehensive and detailed exploration of the scalability of contact-electro-catalytic CO₂RR. Our analysis delves into the shifts in efficiency, cost-effectiveness, and adaptability encountered during the transition from laboratory research to real-world applications of CO₂RR. Furthermore, we scrutinize the influences of reactant concentrations, impurities, and environmental factors, including temperature and humidity, on CO₂RR in practical scenarios. This multifaceted evaluation encompasses not only the overall impact on yield but also the implications for cost-effectiveness. Finally, we conducted an inductive analysis of the challenges encountered in transitioning contact-electro-catalytic CO₂RR from laboratory research to large-scale application, as shown in **Figure R25**. Below, we will break down the impact of reactant concentrations, impurities, and environmental factors on efficiency, adaptability, and cost-effectiveness in laboratory studies, real gas flow conditions, and large-scale applications:

1. Laboratory research:

The Faradaic efficiency in laboratory studies is influenced by the concentration of reactants, impurities, as well as environmental factors such as temperature and humidity. Firstly, lower reactant concentrations diminish the CO₂ adsorption rate, consequently impacting the reaction kinetics and Faradaic efficiencies⁴⁶. Secondly, the presence of impurities, such as oxygen, can compete with CO₂ for active sites on the catalyst,

leading to reduced selectivity and efficiency (**Figure R26a**)⁴⁷. A crucial aspect lies in comprehending and mitigating the influences of impurities to maintain catalyst selectivity and ensure high Faradaic efficiency. Finally, the system temperature will influence reaction rates, while humidity impacts reactant transport and adsorption. Humidity directly affects the electrical output performance of TENG^{25, 39}, where lower humidity generally leads to a decrease in electrical output and thus reduced catalytic efficiency. Therefore, optimizing reaction conditions concerning temperature and humidity is crucial to achieving and maintaining high Faradaic efficiency.

2. Real airflow condition:

2.1. For CO Faradaic efficiency, the introduction of impurities, especially oxygen, has an impact on the Faradaic efficiency of CO₂RR. This is attributed to the competitive nature of the oxygen reduction reaction (ORR) with CO₂RR on the catalyst surface. The presence of impurities in the air, such as oxygen, has been demonstrated to affect the Faradaic efficiency of contact-electro-catalytic CO₂RR. As evidenced in our manuscript (Figure R26b), the Faradaic efficiency was 96.24% under CO₂/Ar conditions, whereas under real airflow conditions, it decreased to approximately 93.95%, indicating an impact from the ORR. Oxygen can occupy active sites, reducing the selectivity and efficiency of CO₂RR. In-depth studies on catalyst surface modification or the development of catalysts with improved selectivity under the presence of these impurities are crucial. Innovative approaches such as tailored catalyst formulations or advanced surface coatings may offer promising solutions. Furthermore, temperature and humidity directly influence the electrochemical kinetics of CO₂RR. Elevated temperatures may lead to a reduction in TENG electrical output due to altered reaction kinetics⁴⁸. Understanding the temperature dependence of the electrochemical processes involved and implementing active temperature control mechanisms, such as cooling systems, is essential. Likewise, low humidity affects both TENG performance and Faradaic efficiency, demanding strategies like humidity control or the use of moisture-retaining materials to ensure consistent and optimal electrochemical performance.

2.2. For adaptability, the adaptability of the CO₂ electroreduction process is contingent

upon its ability to function effectively across varying concentrations of CO₂ and fluctuating levels of impurities. Strategies to enhance adaptability involve the development of catalysts resilient to impurity interference and possibly incorporating gas purification systems. Moreover, investigating the use of alternative catalyst materials with heightened impurity tolerance can further bolster the technology's adaptability. Additionally, adapting the CO₂ electroreduction process to variable temperature and humidity conditions requires a holistic approach. It involves optimizing TENG operation at lower temperatures and higher humidity levels. Design considerations, such as incorporating insulating materials to manage temperature and utilizing humidity-regulating components, play a pivotal role. Furthermore, the development of materials robust to variations in these conditions will be instrumental in achieving sustained adaptability.

2.3. For cost-effectiveness, in the analysis of CO₂RR under real airflow conditions, the focus is predominantly on offering insights for extensive practical implementations. Firstly, the materials used, quaternized CNF and PVDF for TENG preparation, are abundant and can be produced on a large scale. The electrospinning technology for PVDF is progressively maturing⁴⁰, and the one-step firing process for Cu-PCN is straightforward and cost-efficient. Secondly, the TENG movement can be powered by harnessing environmental mechanical energy, endowing the overall system with improved sustainability and cost-effectiveness. However, certain factors may introduce uncertainties in the cost-benefit analysis. Impurities such as oxygen and nitrogen in real gas flows, coupled with variations in temperature and humidity, can influence cost considerations. Oxygen, for instance, might compete for CO₂ sites, potentially reducing CO production. Additionally, variations in temperature and humidity may impact the electrical output of TENG, leading to potential reductions in benefits during large-scale applications.

3. Large-scale application:

In the context of large-scale CO₂RR application, paramount importance should be accorded to cost-effectiveness. While reactant costs become less critical in real gas flow conditions where air is abundant, attention shifts to other cost-related factors.

Integrating efficient gas purification systems to mitigate the effects of impurities becomes crucial. Investing in cost-effective purification methods and technologies, possibly exploring recyclable or regenerable systems, will be central to ensuring economical CO₂ electroreduction. In addition, managing impurities involves a twofold strategy. Firstly, investing in technologies to reduce the presence of impurities in the air stream, possibly through advanced filtration systems, holds potential. Secondly, optimizing catalyst formulations to resist the negative impact of impurities, perhaps through tailored surface modifications or alloying, can enhance cost-effectiveness. At last, cost-effectiveness considerations necessitate the exploration of energy-efficient methods for maintaining low temperature and high humidity in TENG operation. Passive strategies, such as incorporating insulating materials, and active strategies, like efficient cooling and humidity regulation systems, need to be balanced for optimal economic viability. Through a thorough evaluation encompassing efficiency, cost-effectiveness, and adaptability considerations, our contact-electro-catalytic CO₂RR technology demonstrates a rational transition from laboratory research to large-scale applications, laying a robust foundation for practical implementation.

Figure R25. An inductive analysis of the challenges encountered in transitioning contact-electro-catalytic CO₂RR from laboratory research to large-scale application.

Figure R26. (a) Comparison of adsorption energies of CO₂, O₂, and N₂ by quaternized CNF. (b) Comparison of Faradaic efficiency of contact-electro-catalytic CO₂RR under CO₂ and real air flow.

Comment 13: An important aspect to consider in this analysis is how employing a real gaseous stream may impact the results and conclusions. The authors should address potential variations or challenges that could arise when transitioning from synthetic laboratory conditions to real-world scenarios. Discussing the possible differences in reactant concentrations, impurities, and other environmental factors will add a layer of realism to the study and provide a more accurate assessment of the technology's viability outside controlled laboratory settings. This discussion will also contribute to a more well-rounded evaluation of the research's practical implications and potential for real-world applications, similar to the stability of the electrodes.

Authors' reply: We appreciate your valuable insight regarding the potential impact of transitioning from synthetic laboratory conditions to real-world scenarios on our results and conclusions. Acknowledging the importance of this aspect, we have carefully considered the potential variations and challenges that may arise when implementing our technology in practical, real-world settings. We have provided a comprehensive discussion on the progression of contact-electro-catalytic CO₂RR from laboratory studies to real-world scenarios in response to "Comment 12". Our analysis delves into the effects of alterations in reactant concentrations, impurities, and environmental factors on the synthesis efficiency, cost-effectiveness, and adaptability of CO, along with potential solutions. Here, we conduct an analysis of additional factors that could

impact the equipment during the transition from laboratory research to real airflow conditions, as outlined in **Table R4**. Additionally, the TENG exhibits substantial electrical output, robust environmental adaptability, and durability as a new form of clean energy. Previous research has shown that TENG presents distinct advantages in catalyzing hydrogen peroxide production and organic pollutants degradation⁴⁹⁻⁵³. Our laboratory research further affirms the feasibility of TENG for contact-electro-catalytic CO₂ reduction with exceptionally high CO Faradaic efficiency. This work serves as a valuable starting point for converting CO₂ into products with higher added value (e.g., multi-carbon compounds), representing a more promising application prospect for contact-electro-catalytic CO₂RR in the future.

Table R4. Analysis of the factors influencing equipment performance during the transition from laboratory research to real airflow conditions.

Factors	Laboratory research	Real airflow condition
Electrochemical properties	Gas reactants are in a sealed chamber, which does not impact the electrical output performance of the TENG.	Flow rate and diffusion properties of gas reactants affect the electrical output performance of TENG
Device durability	After 35-h of operation, the TENG continues to exhibit a consistent electrical output and maintains efficient catalytic performance.	The presence of impurities in the real gas stream adversely affects the stability of the catalyst, leading to a slight reduction in catalytic performance.
Catalytic properties	The catalyst shows notable activity, with a FE _{CO} exceeding 96%.	Under actual gas flow conditions, impurities and environmental factors cause catalytic selectivity to decrease to 93%.
Mass transfer effect	The gas reactants in the sealed chamber are captured by the electrode material,	Reaction kinetics and mass transfer effects in actual gas flows can lead to changes in efficiency

	yielding CO as the primary product.	and product distribution.
--	-------------------------------------	---------------------------

References

1. Han N, *et al.* Supported Cobalt Polyphthalocyanine for High-Performance Electrocatalytic CO₂ Reduction. *Chem* **3**, 652-664 (2017).
2. Li J, *et al.* Efficient electrocatalytic CO₂ reduction on a three-phase interface. *Nat. Catal.* **1**, 592-600 (2018).
3. Liu M, *et al.* Post-synthetic modification of covalent organic frameworks for CO(2) electroreduction. *Nat. Commun.* **14**, 3800 (2023).
4. Sun X-C, Yuan K, Zhou J-H, Yuan C-Y, Liu H-C, Zhang Y-W. Au³⁺ Species-Induced Interfacial Activation Enhances Metal–Support Interactions for Boosting Electrocatalytic CO₂ Reduction to CO. *ACS Catal.* **12**, 923-934 (2021).
5. Zhuang S, *et al.* Hard-Sphere Random Close-Packed Au(47) Cd(2) (TBBT)(31) Nanoclusters with a Faradaic Efficiency of Up to 96 % for Electrocatalytic CO(2) Reduction to CO. *Angew. Chem. Int. Ed. Engl.* **59**, 3073-3077 (2020).
6. Zhao R, *et al.* Partially Nitrided Ni Nanoclusters Achieve Energy-Efficient Electrocatalytic CO(2) Reduction to CO at Ultralow Overpotential. *Adv. Mater.* **35**, e2205262 (2023).
7. Zou H, *et al.* Quantifying the triboelectric series. *Nat. Commun.* **10**, 1427 (2019).
8. Xu W, *et al.* A droplet-based electricity generator with high instantaneous power density. *Nature* **578**, 392-396 (2020).
9. Wang N, Liu Y, Ye E, Li Z, Wang D. Control methods and applications of interface contact electrification of triboelectric nanogenerators: a review. *Mater. Res. Lett.* **10**, 97-123 (2022).
10. Wang ZL, Wang AC. On the origin of contact-electrification. *Mater. Today* **30**, 34-51 (2019).
11. Kim S, *et al.* Transparent flexible graphene triboelectric nanogenerators. *Adv. Mater.* **26**, 3918-3925 (2014).

12. Kim DW, Lee JH, You I, Kim JK, Jeong U. Adding a stretchable deep-trap interlayer for high-performance stretchable triboelectric nanogenerators. *Nano Energy* **50**, 192-200 (2018).
13. Kim MP, Ahn CW, Lee Y, Kim K, Park J, Ko H. Interfacial polarization-induced high-k polymer dielectric film for high-performance triboelectric devices. *Nano Energy* **82**, 105697 (2021).
14. Tao Z, Pearce AJ, Mayer JM, Wang H. Bridge sites of Au surfaces are active for electrocatalytic CO₂ reduction. *J. Am. Chem. Soc.* **144**, 8641-8648 (2022).
15. Chen S, *et al.* Boosting CO₂-to-CO conversion on a robust single-atom copper decorated carbon catalyst by enhancing intermediate binding strength. *J. Mater. Chem. A* **9**, 1705-1712 (2021).
16. Li Y, Li B, Zhang D, Cheng L, Xiang Q. Crystalline carbon nitride supported copper single atoms for photocatalytic CO₂ reduction with nearly 100% CO selectivity. *ACS nano* **14**, 10552-10561 (2020).
17. Xu C, *et al.* Highly selective two-electron electrocatalytic CO₂ reduction on single-atom Cu catalysts. *Small Struct.* **2**, 2000058 (2021).
18. Hao J, *et al.* Interatomic electron transfer promotes electroreduction CO₂-to-CO efficiency over a CuZn diatomic site. *Nano Res.*, 1-8 (2023).
19. Zhang Q, Guan J. Single-atom catalysts for electrocatalytic applications. *Adv. Funct. Mater.* **30**, 2000768 (2020).
20. Shao Y, Feng C-p, Deng B-w, Yin B, Yang M-b. Facile method to enhance output performance of bacterial cellulose nanofiber based triboelectric nanogenerator by controlling micro-nano structure and dielectric constant. *Nano Energy* **62**, 620-627 (2019).
21. Cheon S, *et al.* High-performance triboelectric nanogenerators based on electrospun polyvinylidene fluoride–silver nanowire composite nanofibers. *Adv. Funct. Mater.* **28**, 1703778 (2018).
22. Fan F-R, Tian Z-Q, Lin Wang Z. Flexible triboelectric generator. *Nano Energy* **1**, 328-334 (2012).
23. Yang W, *et al.* Fundamental research on the effective contact area of micro-/nano-

- textured surface in triboelectric nanogenerator. *Nano Energy* **57**, 41-47 (2019).
24. Zhang R, Olin H. Material choices for triboelectric nanogenerators: A critical review. *EcoMat* **2**, e12062 (2020).
 25. Wang N, *et al.* Dual-electric-polarity augmented cyanoethyl cellulose-based triboelectric nanogenerator with ultra-high triboelectric charge density and enhanced electrical output property at high humidity. *Nano Energy* **103**, 107748 (2022).
 26. Yang D, *et al.* Humidity-Resistant Triboelectric Nanogenerator Based on a Swelling-Resistant and Antiwear PAN/PVA-CaCl₂ Composite Film for Seawater Desalination. *Adv. Funct. Mater.*, 2306702 (2023).
 27. Wang N, *et al.* New Hydrogen Bonding Enhanced Polyvinyl Alcohol Based Self-Charged Medical Mask with Superior Charge Retention and Moisture Resistance Performances. *Adv. Funct. Mater.* **31**, 2009172 (2021).
 28. Cao X, *et al.* Atomic bridging structure of nickel–nitrogen–carbon for highly efficient electrocatalytic reduction of CO₂. *Angew. Chem.* **134**, e202113918 (2022).
 29. Ma W, He X, Wang W, Xie S, Zhang Q, Wang Y. Electrocatalytic reduction of CO₂ and CO to multi-carbon compounds over Cu-based catalysts. *Chem. Soc. Rev.* **50**, 12897-12914 (2021).
 30. Gao Z-H, *et al.* A heteroleptic gold hydride nanocluster for efficient and selective electrocatalytic reduction of CO₂ to CO. *J. Am. Chem. Soc.* **144**, 5258-5262 (2022).
 31. Hooe SL, Dressel JM, Dickie DA, Machan CW. Highly efficient electrocatalytic reduction of CO₂ to CO by a molecular chromium complex. *ACS Catal.* **10**, 1146-1151 (2019).
 32. Liu S, *et al.* Shape-dependent electrocatalytic reduction of CO₂ to CO on triangular silver nanoplates. *J. Am. Chem. Soc.* **139**, 2160-2163 (2017).
 33. Mezzavilla S, Horch S, Stephens IE, Seger B, Chorkendorff I. Structure sensitivity in the electrocatalytic reduction of CO₂ with gold catalysts. *Angew. Chem., Int. Ed.* **58**, 3774-3778 (2019).
 34. Hsu C-S, *et al.* Activating dynamic atomic-configuration for single-site

- electrocatalyst in electrochemical CO₂ reduction. *Nat. Commun.* **14**, 5245 (2023).
35. Ren H, *et al.* Operando proton-transfer-reaction time-of-flight mass spectrometry of carbon dioxide reduction electrocatalysis. *Nat. Catal.* **5**, 1169-1179 (2022).
 36. Gao W, Xu Y, Fu L, Chang X, Xu B. Experimental evidence of distinct sites for CO₂-to-CO and CO conversion on Cu in the electrochemical CO₂ reduction reaction. *Nat. Catal.* **6**, 885-894 (2023).
 37. Chen C, *et al.* Exploration of the bio-analogous asymmetric C–C coupling mechanism in tandem CO₂ electroreduction. *Nat. Catal.* **5**, 878-887 (2022).
 38. Zhang T, Bui JC, Li Z, Bell AT, Weber AZ, Wu J. Highly selective and productive reduction of carbon dioxide to multicarbon products via in situ CO management using segmented tandem electrodes. *Nat. Catal.* **5**, 202-211 (2022).
 39. Wang N, Zheng Y, Feng Y, Zhou F, Wang D. Biofilm material based triboelectric nanogenerator with high output performance in 95% humidity environment. *Nano Energy* **77**, 105088 (2020).
 40. Yang D, *et al.* An asymmetric AC electric field of triboelectric nanogenerator for efficient water/oil emulsion separation. *Nano Energy* **90**, 106641 (2021).
 41. Kim B, *et al.* Dual-Atom-Site Sn-Cu/C₃N₄ Photocatalyst Selectively Produces Formaldehyde from CO₂ Reduction. *Adv. Funct. Mater.*, 2212453 (2023).
 42. Alemany-Molina G, Quílez-Bermejo J, Navlani-García M, Morallón E, Cazorla-Amorós D. Efficient and cost-effective ORR electrocatalysts based on low content transition metals highly dispersed on C₃N₄/super-activated carbon composites. *Carbon* **196**, 378-390 (2022).
 43. Song J, *et al.* Promoting Dinuclear-Type Catalysis in Cu₁–C₃N₄ Single-Atom Catalysts. *Adv. Mater.* **34**, 2204638 (2022).
 44. Xu LH, Liu W, Liu K. Single Atom Environmental Catalysis: Influence of Supports and Coordination Environments. *Adv. Funct. Mater.*, 2304468 (2023).
 45. Yan X, *et al.* Synergy of Cu/C₃N₄ Interface and Cu Nanoparticles Dual Catalytic Regions in Electrolysis of CO to Acetic Acid. *Angew. Chem., Int. Ed.* **62**, e202301507 (2023).
 46. Liu M, *et al.* Enhanced electrocatalytic CO₂ reduction via field-induced reagent

- concentration. *Nature* **537**, 382-386 (2016).
47. Lv JJ, *et al.* Microenvironment engineering for the electrocatalytic CO₂ reduction reaction. *Angew. Chem.* **134**, e202207252 (2022).
 48. Xu C, *et al.* Raising the Working Temperature of a Triboelectric Nanogenerator by Quenching Down Electron Thermionic Emission in Contact-Electrification. *Adv. Mater.* **30**, e1803968 (2018).
 49. Zhao J, Zhang X, Xu J, Tang W, Lin Wang Z, Ru Fan F. Contact-electro-catalysis for Direct Synthesis of H₂ O₂ under Ambient Conditions. *Angew. Chem. Int. Ed. Engl.* **62**, e202300604 (2023).
 50. Wang Z, *et al.* Contact-electro-catalysis for the degradation of organic pollutants using pristine dielectric powders. *Nat. Commun.* **13**, 130 (2022).
 51. Li H, Berbille A, Zhao X, Wang Z, Tang W, Wang ZL. A contact-electro-catalytic cathode recycling method for spent lithium-ion batteries. *Nat. Energy* **8**, 1137-1144 (2023).
 52. Shen S, *et al.* High-Efficiency Wastewater Purification System Based on Coupled Photoelectric-Catalytic Action Provided by Triboelectric Nanogenerator. *Nanomicro. Lett.* **13**, 194 (2021).
 53. Dong X, Wang Z, Berbille A, Zhao X, Tang W, Wang ZL. Investigations on the contact-electro-catalysis under various ultrasonic conditions and using different electrification particles. *Nano Energy* **99**, 107346 (2022).

REVIEWER COMMENTS

Reviewer #1 (Remarks to the Author):

I appreciate the authors' efforts to address the reviewer's suggestions and refining their manuscript. However, upon reviewing the revised version, I have identified some additional points for consideration:

1. While the authors mentioned substituting human body movement with wind energy to propel the TENG's tribolayer, the manuscript lacks experimental details or data supporting this modification. It is crucial to elucidate how the wind speed and direction were measured, as well as how the stability and durability of the TENG were ensured under wind conditions. Moreover, the authors should provide insights into how wind energy impacted the performance of CO₂RR and the resulting CO yield. Additional information and evidence are needed to justify this change and assess its feasibility thoroughly.

2. The authors mentioned that by following their proposed protocol, the CO product could be systematically collected and subsequently utilized for further transformations. However, considering the likely low concentration of CO mixed with unreacted air, downstream transformations might be constrained, making this method impractical for real-world applications. Liquid products such as formate might offer more practical benefits and should be considered as an alternative.

Reviewer #2 (Remarks to the Author):

The authors addressed most of my comments with a good amount of detail. However, they still missed two important points.

for comment 2: I was asking for electrical potential on catalyst surfaces, but not on the TENG surfaces, or PVDF surfaces. Authors need to show how the potential built on on the TENG surface is transferred to the catalyst surface. How the electronic potential on the catalysts surface moves up or down in accordance to TENG potential goes positive or negative. How might electrons accumulate or deplete from the catalyst particles in response to the TENG potential.

for comment 4, I was asking how charges were moved externally between cathode and anode, but the authors only showed how charge is transferred at the interface. This is irrelevant. was the cathode and anode electrically connected? if so, any charges flow in between the catalysts on anode/cathode surfaces during the contact/separation cycle externally, in order to complete the electrochemical reaction charge flow cycle? If this is the case, how would charges get through the insulating PVDF. This was my question.

Further more, in all electrochemical systems, cathode and anode are electrically connected through power source, and the catalyzed reactions are considered as the "load". In this case, I assume TENG is

the power source, but where is the load supposed to be? Authors should make it clear in main figures to show the charge flowing and balance cycle by integrating the TENG charge generating process and the catalytic charge transferring process, rather than just a TENG charge generating cycle in SI, which is well known.

Reviewer #3 (Remarks to the Author):

The authors have thoroughly addressed all the questions raised by the reviewer. Therefore, I recommend the publication of the manuscript in Nature Communications.

Comments and Authors' reply

Reviewer #1 (Remarks to the Author):

Comment: I appreciate the authors' efforts to address the reviewer's suggestions and refining their manuscript. However, upon reviewing the revised version, I have identified some additional points for consideration:

Authors' reply: We appreciate for the reviewer's positive comments. We have revised our manuscript according to reviewer's comments point by point. Here, we list all comments and the corresponding replies as follows:

Comment 1: While the authors mentioned substituting human body movement with wind energy to propel the TENG's tribolayer, the manuscript lacks experimental details or data supporting this modification. It is crucial to elucidate how the wind speed and direction were measured, as well as how the stability and durability of the TENG were ensured under wind conditions. Moreover, the authors should provide insights into how wind energy impacted the performance of CO₂RR and the resulting CO yield. Additional information and evidence are needed to justify this change and assess its feasibility thoroughly.

Authors' reply: We thank the reviewer for the insightful comments. As advised by the reviewer, we have designed new experiments to investigate the wind-driven TENG catalyzed CO₂RR and included the new data in the revised manuscript. In the present design, the catalytic system is achieved by integrating a fan and the wind-driven TENG device in a closed reactor, as shown in **Figure R1a** and **b**. The fan was utilized to provide wind energy for TENG movement and was equipped with three speed modes (low, medium, and high). The wind speeds were measured using an anemometer (UT363) and determined to be 4 m/s, 5.8 m/s, and 8 m/s for the fan's low, medium, and high-speed modes, respectively (**Figure R1c-f**). As a proof-of-concept, the wind direction was adjusted to be parallel to the surface of the TENG's tribolayer. During the operation, the quaternized CNF film moves up and down under the influence of the

wind, making contact with the Cu-PCN@PVDF tribolayer. The real-time electrical output of the device was monitored by a computer and used to calculate the CO Faradaic efficiency.

To ensure the stability of the wind-driven TENG device, we reduced the distance between the upper and lower Cu-PCN@PVDF/Al electrodes to 3 cm, facilitating smoother contact between the two tribolayers, *i.e.* Cu-PCN@PVDF and quaternized CNF. Furthermore, we increased the thickness of the quaternized CNF film to 0.3 mm, bestowing the film with enhanced flexibility and potentially reducing the risk of breakage during operation. We then conducted a test on the real-time electrical output of the wind-driven TENG device over 14 h at a wind speed of 8 m/s. As shown in **Figure R1g**, the electrical output of the TENG device remains relatively stable without significant attenuation. The maximum output current after charge accumulation is maintained at approximately 16 μA , indicating the high stability and durability of the TENG device.

To test the CO₂ reduction performance of the wind-driven TENG device, we carried out contact-electro-catalytic CO₂RR experiments under different wind speeds (4 m/s, 5.8 m/s, and 8 m/s). Prior to each test, the fan was activated and the reactor was purged with compressed air to remove any potential impurities. Then the reactor was sealed and the humidity inside was stabilized at 99% RH. The reaction lasted for 5 h after the TENG's electrical output was stabilized. The real-time electrical outputs of the wind-driven TENG are shown in **Figure R1h-j**. The results indicate a gradual increase in the electrical output of the TENG with higher wind speeds, aligning with the greater vibration amplitude and frequency of the TENG tribolayer under these conditions. Afterward, CO production was detected by GC, and the CO Faradaic efficiency (FE_{CO}) was calculated using the same method described in our original manuscript. The FE_{CO} values obtained at wind speeds of 4 m/s, 5.8 m/s, and 8 m/s were determined to be 91.86%, 92.10%, and 92.33%, respectively (**Figure R1k**). These values are comparable to the FE_{CO} (93.95%) obtained for the motor-driven TENG device as reported in our original manuscript. The slight variation in FE_{CO} can be attributed to the influence of

the gas flow on the CO₂ adsorption and product desorption process. To align with these new experimental findings and increase the readability of the manuscript, Figure R1 has been incorporated into the revised manuscript as Figure S13, with the corresponding description highlighted in red color. Furthermore, Figure 4a from the original text has been included in the revised manuscript as Figure 4h, with the corresponding figure caption changes highlighted in red.

Figure R1. (a) Photograph of the contact-electro-catalytic CO₂ reduction system based on the wind-driven TENG device. (b) Front view and side view (inset) of the wind-driven TENG device. (c) Photograph showing the measurement of fan wind speed using an anemometer (UT363). (d-f) Wind speeds measured at different fan speed modes, representing the low (4 m/s) (d), medium (5.8 m/s) (e), and high-speed wind (8 m/s) (f). (g) Stability test for the wind-driven TENG device at a wind speed of 8 m/s. (h-j) Electrical output of the wind-driven TENG for contact-electro-catalytic CO₂RR at wind speeds of 4 m/s (h), 5.8 m/s (i), and 8 m/s (j). (k) Comparison of CO Faradaic efficiencies obtained using the wind-driven TENG device under different wind speeds.

Comment 2: The authors mentioned that by following their proposed protocol, the CO product could be systematically collected and subsequently utilized for further transformations. However, considering the likely low concentration of CO mixed with unreacted air, downstream transformations might be constrained, making this method impractical for real-world applications. Liquid products such as formic acid might offer more practical benefits and should be considered as an alternative.

Authors' reply: We thank the reviewer for the constructive comments. We agree with the reviewer on the comment that liquid products such as formic acid, acetic acid, and methanol are easier to collect and may enhance the practicality of the current catalytic system for real-world applications. In this study, we would like to emphasize that our primary goal is to demonstrate the feasibility and new perspective of using contact-electro-catalysis as a novel approach for CO₂ conversion, which is supported by both experimental findings and theoretical simulations reported in the manuscript. While we have successfully demonstrated the proof-of-concept of contact-electro-catalytic CO₂ reduction to CO from ambient air, we agree with the reviewer that, as previously stated, this method still faces challenges in real-world applications due to the low concentration of CO and its complex mixture with unreacted air, among other factors. However, it is worth noting that the contact-electro-catalytic system developed in this work has the potential to serve as a novel catalytic platform for a seamless integration of other different emerging single-atom catalysts. This allows us to fine-tune the catalyst type and the number of catalytic sites in TENG, aiming to achieve excellent selectivity in contact-electro-catalytic CO₂RR for liquid product generation. We will include the findings in a future detailed report on this topic. In the current revision, we have included new information to discuss its potential in this regard, showing as follows: "Moreover, in future investigations, through the optimization of catalyst type and the number of catalytic sites, contact-electro-catalytic CO₂RR holds the potential to generate products of higher value than CO, particularly in the realm of liquid products", as highlighted in red in the Discussion section of the revised manuscript.

Reviewer #2 (Remarks to the Author):

Comment: The authors addressed most of my comments with a good amount of detail. However, they still missed two important points.

Authors' reply: We appreciate for the reviewer's comments. First of all, we apologize for the misunderstanding of the reviewer's comments in the first round of revision and the unclear responses regarding the "potential establishment on the catalyst surface" and "charge transfer between cathode and anode". Here, we list all comments and the corresponding replies as follows:

Comment 1: for comment 2, I was asking for electrical potential on catalyst surfaces, but not on the TENG surfaces, or PVDF surfaces. Authors need to show how the potential built on the TENG surface is transferred to the catalyst surface. How the electronic potential on the catalysts surface moves up or down in accordance to TENG potential goes positive or negative. How might electrons accumulate or deplete from the catalyst particles in response to the TENG potential.

Authors' reply: We thank the reviewer for the valuable comments. We apologize for the oversight in not including the discussion on the establishment of the catalyst surface potential in the previous response. During the triboelectric charging process of quaternized CNF and PVDF, the triboelectrons temporarily stored on the PVDF surface can move freely within a certain range (e.g., charge dissipation)^{1,2}, which offers the possibility for electron transfer from PVDF to Cu-PCN, enabling the establishment of electric potential on the catalyst surface. To further demonstrate the electron transfer between PVDF and Cu-PCN, we employed DFT calculations³⁻⁵ to investigate the charge distribution at the interface of these two components. As shown in **Figure R2**, yellow and cyan regions indicate electron accumulation and depletion, respectively. The interaction between PVDF and Cu-PCN leads to obvious electron accumulation on Cu-PCN, especially on the Cu single atoms. This result suggests that the triboelectrons can be transferred from the TENG's tribolayer to the Cu-PCN, enabling the establishment of the catalyst surface potential and the subsequent CO₂ reduction.

Figure R2. (a) Simulated charge distribution at the interface of PVDF and Cu-PCN. (b) side view and (c) top view of the structural model utilized to simulate the interaction between PVDF and Cu-PCN. The blue, white, purple, pink, and red balls represent N, C, F, H, and Cu atoms, respectively. Yellow and cyan regions indicate electron accumulation and depletion, respectively.

To provide a clearer understanding of the correlation between the electrical potential on the catalyst surface and the electrical output of the TENG during contact-electrocatalytic CO₂RR, we integrated the TENG charge generation process with the catalytic charge consumption process into a single TENG working cycle, as shown in **Figure R3**. As depicted in **Figure R3i**, the TENG device comprises two components, an electronegative tribolayer on the Aluminium (Al) back electrode (Cu-PCN@PVDF/Al) and an electropositive tribolayer on the Al back electrode (CNF/Al). The two Al back electrodes are connected via an external circuit with an ammeter attached to measure the electrical output of the TENG. The initial stage of contact electrification involves the complete contact of the two tribolayers, leading to the generation of triboelectric charges. It is worth noting that the charges located on the tribolayer surface cannot penetrate the interior of tribolayers (Cu-PCN@PVDF and quaternized CNF) to reach the Al back electrodes positioned behind them, owing to the insulating nature of PVDF and CNF. Instead, the electrostatic induction between the tribolayers and back electrodes results in the deposition of induced charges on the two Al back electrodes, accompanied by the current generation in the external circuit (i.e., TENG's electrical

output)⁶. When the two tribolayers are separated, induced electrons migrate from the Al back electrode of Cu-PCN@PVDF/Al to that of CNF/Al through the external circuit, indicating a current direction pointing from CNF/Al to Cu-PCN@PVDF/Al. Furthermore, CO₂ molecules can be captured by the quaternized CNF and interact with electrons stored in the Cu-PCN catalyst to initiate CO₂ reduction (**Figure R3ii**). In the meantime, water molecules in the high-humidity environment undergo oxidation on the surface of positively charged quaternized CNF. When the separation distance between the two tribolayers reaches the maximum, the two electrodes reach an electrical equilibrium state and no current flow (i.e., TENG's electrical output) can be detected in the external circuit (**Figure R3iii**). At this point, the induced charge on the two Al back electrodes also reaches its maximum level. Afterward, the Cu-PCN@PVDF/Al electrode is compressed downward, causing the induced electrons on the Al back electrode of CNF/Al to flow back toward that of the Cu-PCN@PVDF/Al (**Figure R3iv**). As the compression process advances, CO₂ molecules adsorbed on the quaternized CNF surface come into contact with Cu-PCN, facilitating the electron transfer for CO₂ reduction (**Figure R3v**). Concurrently, water molecules undergo oxidation to complete the catalytic cycle. Subsequently, the charge in the tribolayers of the TENG is exhausted, and the two electrodes reach another equilibrium state with no current flow in the external circuit (**Figure R3vi**). The electrical output of the TENG initiates the next identical working cycle once the two tribolayers make complete contact and triboelectric charges are generated. Throughout the TENG working cycle, the quantity of triboelectric charges participating in the catalytic reaction equals the quantity of the **induced charges** flowing through the external circuit via the two back electrodes.

Based on the analysis of **Figures R2** and **R3**, contact electrification and electron transfer establish an electric potential on the Cu-PCN surface. The catalyst surface potential reaches its maximum at the beginning of the TENG working cycle (i.e., contact electrification). Subsequently, this potential gradually decreases as the triboelectric charges participate in surface CO₂ conversion. This phenomenon contrasts sharply with the observed TENG output current (the induced current in the external

circuit), which changes direction during the tribolayer contact and separation process (**Figure R3**).

To prevent any potential misunderstanding regarding the difference between the **potential on the catalyst surface** and the **induced potential in the external circuit**, we further provide below a more detailed explanation of the induced current of TENG. The theoretical foundation of TENG is rooted in Maxwell's displacement current, which arises from a changing electric field and differs from the conduction current generated by free electrons⁷. During TENG operation, the contact and separation of tribolayers leads to variations in the electric field. The motion state of the TENG's tribolayer changes due to external forces, accompanied by contact electrification and charge generation on the surface of tribolayers. This phenomenon not only results in temporal changes in the local charge density ρ on the tribolayer's surface but also induces local "virtual" current density, leading to dielectric polarization⁸⁻¹⁰. According to Maxwell's equations, the expression for the electric displacement vector is¹¹:

$$D = D' + P_s = \varepsilon E + P + P_s \quad \text{Eq (1)}$$

In the formula, εE is the electric field caused by free charges, where ε is the vacuum dielectric constant; P represents the polarization inside the medium (induced polarization); P_s represents dynamic rising polarization. According to Eq (1), the expression of the displacement current J_D is:

$$J_D = \frac{\partial D'}{\partial t} + \frac{\partial P_s}{\partial t} \quad \text{Eq (2)}$$

In the formula, $\frac{\partial D'}{\partial t}$ represents the displacement current caused by the time-varying electric field, which is called the induced displacement current; $\frac{\partial P_s}{\partial t}$ represents the current generated by the charged medium under the action of external force, which is called the kinetic displacement current. Due to contact electrification, the surfaces of the two tribolayers of TENG acquire opposite charges, with the surface charge density defined as $\pm\sigma_{tribo}$ (typically reaching saturation after initial contact, regardless of the separation distance z). The electrostatic charge induced by friction on the surface generates an electric field, which propels the movement of the tribolayer. Free electrons

from the metal back electrode flow through the external circuit, and the amount of transferred charge $\pm\sigma_{tribo}$ accumulated on the electrode depends on z . Consequently, the mechanical energy causing the change in z is converted into electrical energy. The corresponding displacement current is calculated as follows:

$$J_{D_z} = \frac{\partial D_z}{\partial t} = \frac{\partial D_{tribo}(z,t)}{\partial t} \quad \text{Eq (3)}$$

In the formula, D_z is the electric displacement vector when the separation distance is z ; $\frac{\partial D_z}{\partial t}$ is the displacement current caused by the time-varying electric field; $\frac{\partial D_{tribo}(z,t)}{\partial t}$ is the time differential of the transferred charge. Therefore, the induced current measured by the external circuit is the displacement current resulting from the time-varying electric field, and its change in direction is due to alterations in the movement direction of the TENG's tribolayer. Returning to the previous question, the induced electricity measured in the TENG's external circuit can indicate the number of triboelectric charges involved in CO₂ conversion, despite the disparity between the potential on the catalyst surface and the TENG output potential (the induced potential). **Figure R2** has been added to the revised manuscript as **Figure S3**; **Figure R3** has been added to the revised manuscript as **Figure 1b**, and the corresponding description is marked in red.

Figure R3. Schematic diagram of the TENG working cycle, highlighting the integrated TENG charge generation and catalytic charge consumption process during contact-electro-catalytic CO₂RR. The charges generated in the TENG device include both triboelectric charges on the tribolayer surface and induced charges on metal back electrodes.

Comment 2: for comment 4, I was asking how charges were moved externally between cathode and anode, but the authors only showed how charge is transferred at the interface. This is irrelevant. was the cathode and anode electrically connected? if so, any charges flow in between the catalysts on anode/cathode surfaces during the contact/separation cycle externally, in order to complete the electrochemical reaction charge flow cycle? If this is the case, how would charges get through the insulating PVDF. This was my question.

Authors' reply: We thank the reviewer for highlighting the concern. We apologize for not explaining clearly how the charge moves externally between the cathode and anode in our previous reply and wish to clarify as following. In a traditional electrocatalysis device setting, it requires a charge flow cycle to complete the electrochemical reaction (**Figure R4a**). However, in our contact-electro-catalytic system, the TENG device comprises two components, an electronegative tribolayer on the Al back electrode (Cu-PCN@PVDF/Al) and an electropositive tribolayer on the Al back electrode (CNF/Al). And the two Al back electrodes are connected via an external circuit with an ammeter attached to measure the electrical output of the TENG. We would like to emphasize that the electrons flowing through the external circuit of the TENG are the induced charges on the Al back electrodes rather than the direct triboelectric charges on the surface of tribolayers. In detail, the classical theory of triboelectrification posits that the electrons residing on the electronegative tribolayer are unable to revert back to the electropositive tribolayer, even when the two tribolayers are in full contact, owing to the potential difference between the two tribolayers¹². Therefore, the triboelectrons on the Cu-PCN@PVDF surface cannot shuttle freely to the quaternized CNF during contact between the two tribolayers (**Figure R4ii**). Additionally, the electron migration between the two separated tribolayers as well as between the tribolayer surfaces and the Al back electrodes is prohibited due to the insulating properties of air and triboelectric materials (PVDF and CNF) (**Figure R4i and iii**). Instead, the electrostatic induction between the tribolayers and the back electrodes results in the deposition of induced charges on the two Al back electrodes, accompanied by the current generation in the external circuit

(i.e., TENG's electrical output).

It is worth noting that the triboelectrons stored on the tribolayer surface are able to transfer to the catalyst surface and participate directly in CO₂ conversion upon contact between the catalyst and CO₂ reactants. This has been explained in the response to comment 1 as mentioned above and the phenomenon is consistent with the recent findings reported by other groups. As an example highlighted here, the physical contact between polytetrafluoroethylene particles and deionized water/O₂ under the action of mechanical force can directly lead to charge redistribution and water/O₂ activation for free radical generation¹³, as shown in **Figure R5a**. Similarly, the interaction between SiO₂ and H₂O results in electron transfer between the two components and the subsequent molecule activation¹⁴ (**Figure R5b**). In these studies, the triboelectrons stored in polytetrafluoroethylene or SiO₂ directly participate in surface catalysis without the necessity for a charge flow cycle (or an external circuit). In the current work, the external circuit is integrated with the TENG device to quantify the induced charges, thus determining the number of triboelectric charges used for CO₂RR. We have added a description in the "Introduction" section of the revised manuscript, clarifying that in contact-electro-catalysis, triboelectrons can directly participate in surface catalysis without the necessity for a charge flow cycle. The corresponding modifications are marked in red. We hope the above explanation can help to clarify the reviewer's concern on this point.

Figure R4. (a) Schematic diagram of the charge flow cycle in a traditional electrocatalytic system. (b) Schematic diagram of the electron flow in contact-electrocatalysis. Due to the insulating properties of air and the tribolayer materials (PVDF and quaternized CNF), electron transfer between the two separated tribolayers and between the tribolayer surfaces and the Al back electrodes is prohibited.

Figure R5. (a) Previous report showing the proposed mechanism for contact-electrocatalytic H_2O_2 production from H_2O and O_2 molecules¹³. Copyright 2023, Wiley-VCH. (b) Previous report showing the proposed mechanism for contact-electro-catalytic molecule activation, indicating the direct participation of triboelectric charges in catalysis without the necessity for an external circuit and a charge flow cycle¹⁴. Copyright 2023, Springer Nature.

Comment 3: Furthermore, in all electrochemical systems, cathode and anode are electrically connected through power source, and the catalyzed reactions are considered as the "load". In this case, I assume TENG is the power source, but where is the load supposed to be? Authors should make it clear in main figures to show the charge flowing and balance cycle by integrating the TENG charge generating process and the catalytic charge transferring process, rather than just a TENG charge generating cycle in SI, which is well known.

Authors' reply: We thank the reviewer for the valuable comments. We apologize for not clearly describing the TENG charge generation process and the catalytic charge consumption process in our previous revision. As explained in the above-mentioned response to comments 1 and 2, contact-electro-catalysis, unlike the traditional electrocatalysis, takes place inside the TENG, and there is no catalytic reaction occurring in the external circuit. The external circuit is integrated with the TENG device to quantify the induced charges. However, the triboelectrons stored on the catalyst surface can directly participate in catalysis without the necessity for an external circuit. To clarify on this point, we have integrated the TENG charge generation process and catalytic charge consumption process into one figure (**Figure R6**), which is now included in the revised manuscript as **Figure 1b**.

Figure R6. (a) Schematic diagram of the TENG structure and the contact-electrocatalytic CO₂ reduction process. (b) Schematic diagram of the TENG working cycle, highlighting the integrated TENG charge generation and catalytic charge consumption process during contact-electrocatalytic CO₂RR. The charges generated in the TENG device include both triboelectric charges on the tribolayer surface and the induced charges on the metal back electrodes.

References

1. Wang ZL, Wang AC. On the origin of contact-electrification. *Mater. Today* **30**, 34-51 (2019).
2. Wang N, Pu M, Ma Z, Feng Y, Guo Y, Guo W, Zheng Y, Zhang L, Wang Z, Feng M. Control of triboelectricity by mechanoluminescence in ZnS/Mn-containing polymer films. *Nano Energy* **90**, 106646 (2021).
3. Kresse G, Furthmüller J. Efficient iterative schemes for ab initio total-energy calculations using a plane-wave basis set. *Phys. Rev. B* **54**, 11169 (1996).
4. Perdew JP, Burke K, Ernzerhof M. Generalized gradient approximation made simple. *Phys. rev. lett.* **77**, 3865 (1996).
5. Blöchl PE. Projector augmented-wave method. *Phys. Rev. B* **50**, 17953 (1994).
6. Niu S, Wang ZL. Theoretical systems of triboelectric nanogenerators. *Nano Energy* **14**, 161-192 (2015).
7. Wang ZL. Triboelectric nanogenerators as new energy technology for self-powered systems and as active mechanical and chemical sensors. *ACS nano* **7**, 9533-9557 (2013).
8. Li S, Deng S, Xu R, Liu D, Nan Y, Zhang Z, Gao Y, Lv H, Li M, Zhang Q. High-frequency mechanical energy harvester with direct current output from chemical potential difference. *ACS Energy Lett.* **7**, 3080-3086 (2022).
9. Wang Z, An J, Nie J, Luo J, Shao J, Jiang T, Chen B, Tang W, Wang ZL. A self-powered angle sensor at nanoradian-resolution for robotic arms and personalized medicare. *Adv.Mater.* **32**, 2001466 (2020).
10. Sun W, Li B, Zhang F, Fang C, Lu Y, Gao X, Cao C, Chen G, Zhang C, Wang ZL. TENG-Bot: Triboelectric nanogenerator powered soft robot made of uni-directional dielectric elastomer. *Nano Energy* **85**, 106012 (2021).
11. Wang ZL. Triboelectric nanogenerator (TENG)—sparking an energy and sensor revolution. *Adva. Energy Mater.* **10**, 2000137 (2020).
12. Wang ZL. On Maxwell's displacement current for energy and sensors: the origin of nanogenerators. *Mater. today* **20**, 74-82 (2017).

13. Zhao J, Zhang X, Xu J, Tang W, Lin Wang Z, Ru Fan F. Contact-electro-catalysis for Direct Synthesis of H₂O₂ under Ambient Conditions. *Angew. Chem.* **135**, e202300604 (2023).
14. Li H, Berbille A, Zhao X, Wang Z, Tang W, Wang ZL. A contact-electro-catalytic cathode recycling method for spent lithium-ion batteries. *Nat. Energy* **8**, 1137-1144 (2023).

REVIEWERS' COMMENTS

Reviewer #1 (Remarks to the Author):

I am pleased to acknowledge the improvements made to the revised manuscript following the initial round of peer review. The additional experimental details provided have significantly enhanced the manuscript, offering a clearer understanding of this approach taken in substituting human body movement with wind energy for the TENG's tribolayer propulsion. Additionally, the clarification of how the electrical potential is transferred from the TENG surface to the catalyst surface, along with the detailed explanation of the electron dynamics in response to the TENG potential, appear to have satisfactorily addressed the concerns previously raised by referee 2. Therefore, I recommend the acceptance of this work for publication.

Reviewer #2 (Remarks to the Author):

The authors' answers to my remaining comments are acceptable. I don't have any further comments.

Reviewer #3 (Remarks to the Author):

The authors have thoroughly addressed all the questions raised by the reviewer. Therefore, I recommend the publication of the manuscript in Nature Communications.